# INSPIRE standards as framework for artificial intelligence applications: a landslide example

Gioachino Roberti[1], Jacob McGregor[1], Sharon Lam[1], David Bigelow[1], Blake Boyko[1], Chris Ahern[1], Victoria Wang[1], Bryan Barnhart[1], Clinton Smyth[1], David Poole[1,2], and Stephen Richard[1]

[1]Minerva Intelligence Inc., 301 – 850 West Hastings Street, Vancouver, British Columbia, Canada V6C 1E1
[2]Computer Science department, University of British Columbia, Vancouver, V6T 1Z4, Canada

**Correspondence:** Gioachino Roberti (groberti@minervaintelligence.com)

**Abstract.** This study presents a landslide susceptibility map using an artificial intelligence (AI) approach based on standards set by the INSPIRE framework. INSPIRE is a European Union Spatial Data Infrastructure (SDI) initiative to standardize spatial data across borders to ensure interoperability for management of cross-border infrastructure and environmental issues. However, despite the theoretical effectiveness of the SDI, few real-world applications make use of INSPIRE standards. In this study, we show how INSPIRE standards enhance the interoperability of geospatial data, and enable deeper knowledge development for their interpretation and explainability in AI applications. We designed an ontology of landslides, embedded with INSPIRE vocabularies and then aligned geology, stream network and land cover data sets covering the Veneto region of Italy to the standards. INSPIRE was formally extended to include an extensive landslide type code list, a landslide size code list and the concept of landslide susceptibility to describe map application inputs and outputs. Using the terms in the ontology, we defined conceptual scientific models of areas likely to generate different type of landslides as well as map polygons representing the land surface. Both landslide models and map polygons were encoded as semantic networks and, by qualitative probabilistic comparison between the two, a similarity score was assigned. The score was then used as a proxy for landslide susceptibility and displayed in web map application. The use of INSPIRE-standardized vocabularies in ontologies that express scientific models promotes the adoption of the standards across the European Union and globally. Further, this application facilitates explaining the generated results. We conclude that public and private organisations, within and outside the European Union, can enhance the value of their data by bringing them into INSPIRE-compliance for use in AI applications.

# 1 Introduction

 ## 1.1 INSPIRE

Data accessibility and interoperability is key for multinational cross-border applications and fundamental for economic development (European Parliament and the Council, 2007). Different countries have different languages and data standards, hindering infrastructure planning, disaster risk reduction initiatives, and effective legislative implementation. To overcome these challenges, the European Union initiated INSPIRE (Infrastructure for Spatial Information in the European Community - Directive 2007/2/EC) (European Parliament and the Council, 2007). INSPIRE is structured in 34 spatial data themes organized in three annexes. The themes span administrative (e. g. street addresses) and environmental domains (e. g. geology), and all EU countries are mandated by law to have implemented the data framework by 2021 (European Parliament and the Council, 2014). Each theme defines a data model and has adopted a set of vocabularies to populate interoperable datasets based on that data model. EU countries are aligning and serving INSPIRE data at a slow pace, and currently relatively few INSPIRE-compliant data sets are available across Europe (Cho and Crompvoets, 2019). Conferences and competitions are currently being organized to promote its implementation and to show the potential impact of real-world applications built on INSPIRE data sets (European Commission, 2019). This project was first presented at one of these conferences, the Helsinki 2019 INSPIRE data challenge under the "Let's make the most out of INSPIRE!" topic, where the project won first prize.

## 1.2 Artificial intelligence

Artificial Intelligence (AI) studies "the synthesis and analysis of computational agents that act intelligently" (Poole and Mackworth, 2017). Part of acting intelligently is building models of the world that make predictions. Probabilistic predictions are the most useful ones for subsequent decision making, and can be learned from data (Pearl, 1988). All models are based on human knowledge and data (observations of the world). For some problem domains, society has collected an overwhelming amount of data and still, useful human knowledge of the domain can be very vague. Machine learning has made great progress recently for such cases, particularly with deep learning (Goodfellow et al., 2016). However, for domains with relatively limited, but still very large in volume, data, human knowledge (which may be represented in computer through the use of ontologies) can complement the data to make useful predictions (Pearl, 1988). Many environmental problems do not have enough data (e.g. lack of extensive landslide databases) to be solved by deep learning, but do have enough data to generate useful products when combined with human expertise (expressed in ontologies) (Poole and Mackworth, 2017). The term Artificial Intelligence is commonly used to indicate only the machine learning part of the field, especially in the landslide literature (e.g., Dieu and Gjermundsen, 2020). In this paper we use the term "AI"in its broader connotation, which includes also the ontological method used in this paper. See below for the description of the method and definition of ontologies.

## 1.3 The need for standards, ontologies, and taxonomies

Consistent, well defined vocabularies and data standards are essential in computer science applications, especially in AI. For data to have meaning, and to combine multiple datasets, vocabularies must be consistent and clearly-defined. Deep learning techniques require meanings for the inputs and the outputs (commonly specified in data storage standards such as JPEG, or WAV), but the internal representations do not have well-defined meanings, making the models very opaque (Marcus, 2018). Other representations, such as logical and probabilistic representations, support internal reasoning using symbols with well-defined meanings, which lend themselves to use in explanations (Marcus and Davis, 2019).

Ontologies are "a specification of the meanings of the symbols in an information system" (Poole and Mackworth, 2017). In particular, an ontology defines the vocabulary for individuals and relationships within a knowledge domain. Individuals may be concrete entities (e.g. a rock), or abstract concepts, (e.g. numbers); relationships are properties that describe how individuals are connected. Typical examples of relationships include: is-a-kind-of, is-part-of, is-superclass-of, has-some-property; the ontology also defines axioms controlling the use of the vocabulary for logical and thematic consistency (Poole and Mackworth, 2017). Given these axioms, the vocabulary can be unambiguously interpreted according to the rules of symbolic logic, and implicit relationships between entities or instances of those entities can be inferred.

Vocabularies can be Aristotelian taxonomies, which are logically-consistent and multi-hierarchical. Aristotelian taxonomies are constructed by defining concepts from their relation to a more general parent concept (genus) and using differentiating properties (differentia) to distinguish concepts within the same genus (Aristotle, 350BC). For example, "Slides in soil" and "Slides in rock" share the same parent concept "Slides" and they are differentiated by the property dealing with the material type, "Soil" and "Rock", which make them uniquely identifiable. Taxonomies based on Aristotelian definitions support multi-hierarchical knowledge networks and can be used by computers to make logical inferences (Poole et al., 2009; Smith, 2003). The term 'multi-hierarchical' implies that there is more than one way to move through a taxonomy to arrive at a particular node or term. For example, the landslide taxonomy can be arranged based on different properties. If the landslide types are firstly arranged based on the type of movement and then based on the type of material, one path within the taxonomy would be: Landslide> slides> slides in rock and slides in soil. Alternatively, if the landslide types are arranged first based on the material type and then on the movement type, the path of the taxonomy would be: Landslide> landslides in rock> slides in rock and flows in rock. Both paths are valid, but they reach the same concept in different ways. The Natural Hazard Classification code list extension for landslides presented in this paper was prepared using the open access Aristotelian Class Editor (ACE) software (Minerva Intelligence, 2019d). Knowledge stored in a domain-specific ontology (e.g. geohazards) can be accessed by computers, allowing for data investigation through various artificial intelligence (AI) techniques, including probabilistic matching between semantic networks, the technique used in this study.

Significant progress has been made in the development of taxonomies for geoscience information interchange by the IUGS CGI Geoscience Terminology Working Group which produced the GeoSciML standard along with the OGC (CGI, 2003). However, ontology applications in Earth Sciences are scarce. Notable exceptions are in economic geology (Smyth et al., 2007), geohazards (Jackson Jr et al., 2008), and disaster risk reduction domains (Phengsuwan et al., 2019; Sermet and Demir, 2019).

The INSPIRE framework, through its standardised vocabularies (code lists), provides a necessary foundation upon which AI applications with explainable output can be constructed. INSPIRE application examples in landslide studies include the LAND-deFeND Italian landslide database structure (Napolitano et al., 2018) and a deep learning algorithm to map landslide susceptibility (Hajimoradlou et al., 2020). In Hajimoradlou et al. (2020)'s implementation of deep learning, training features were labelled with INSPIRE-compliant semantics to enable reproducibility of the experiment by other researchers.

In this study, we present an AI-based landslide susceptibility application using a natural hazard ontology. We do so by building from the ontology created by Jackson Jr et al. (2008), and by embedding INSPIRE code lists wherever possible and by aligning input and output data to the INSPIRE data standards.

## 1.4 Landslide susceptibility and hazard

Landslide susceptibility is defined as the relative spatial probability of occurrence for a landslide based on the intrinsic properties of a site (SafeLand, 2011). The concept of susceptibility differs from hazard in that the temporal probability of occurrence, the triggering factors, and the magnitude of the event are not considered in the definition of a susceptibility map (SafeLand, 2011; Van Den Eeckhaut and Hervás, 2012). To produce landslide susceptibility maps, three approaches are usually applied: statistical, physical, and expert-based (SafeLand, 2011). Statistical methods rely on the analysis of landslide databases and their relation to landscape properties (see review by Reichenbach et al., 2018); physical methods calculate the limit equilibrium between failure-resisting and -driving forces in slopes (e.g., Baum et al., 2008); and expert-based methods rely on expert opinion and the assumption that influencing factors are known and are specified in the models (Dai et al., 2002). The AI approach used in this study is an example of the expert-based approach, as the models follow rules that represent the reasoning process of a landslide-expert, providing semi-quantitative susceptibility maps.

## 2 Methods

Figure 1 outlines the methodological workflow followed in this study to produce explainable landslide susceptibility assessments in the Veneto Region of Italy. We extended INSPIRE (Section 2.1), we constructed an ontology (Section 2.2) and we defined expert-models (Section 2.2.1) and instances, represented by mapping polygons (Section 2.2.2). We then compared the similarity of models and instances to produce a matching score, which is used as susceptibility indicator (Section 2.2.3). Finally, the results are delivered in an interactive webmap (Section 2.2.4).

### 2.1 INSPIRE extension

Technical guideline documents outline the data structure for each theme within the INSPIRE directive, its encoding rules, its metadata standards, and some of its use cases. Data structures are formally represented using Unified Modeling Language (UML), modeling thematic entities as feature types, defining properties for each feature type, and characterizing relationships between feature types. Where applicable, standardised vocabularies are adopted for property value ranges. INSPIRE themes can be understood as an ontology (See Section 2.2 below), by defining various entities and the relationships between them.

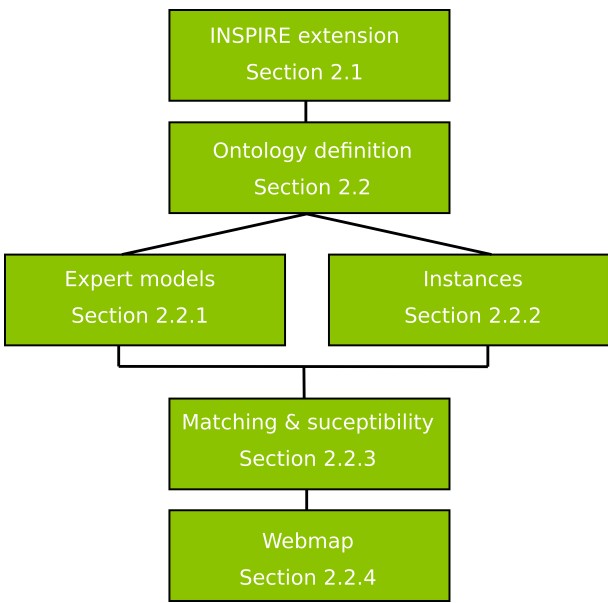

**Figure 1.** The workflow followed in this study and corresponding method sections. We extended INSPIRE, defined and ontology, expert models and mapping instances. We compared models and instances to deliver a susceptibility map which is available online

INSPIRE data models are implemented as Geography Markup Language (GML) application schemas and serialized using Extensible Markup Language (XML). This enables data distribution provided as Open Geospatial Consortium (OGC) - com-
115 pliant web services. Geospatial features are located using vector-based spatial data. Feature properties have value types (e.g. geometry for vector data sets); properties whose value ranges are controlled vocabularies have values implemented as code lists. Code lists incorporate vocabularies developed outside of INSPIRE (e.g. IUGS CGI rock type taxonomy). Some code lists within INSPIRE are not extensible, some are extensible with narrower values, and some allow additional values at any level. Code list values, definitions and hierarchical structures are stored in the INSPIRE registry, making them accessible to
120 and reusable by anyone. INSPIRE schemas can also be extended to include additional concepts and/or feature types. For this project, we worked with four INSPIRE themes: Geology, Land Cover, Hydrography and Natural Risk Zones. The Natural Risk Zone application schema was not fully adequate for this application as it lacked the 'landslide susceptibility' concept and 'landslide type' code lists (Tomas et al., 2015). We addressed this issue by formally extending the Natural Risk Zone schema and the Natural Hazards code list.

## 2.2 Ontologically-grounded probabilistic matching

The method used to produce INSPIRE-based landslide susceptibility maps, uses qualitative probabilistic reasoning that incorporates expert knowledge, making qualitative predictions based on comparisons between models and instances (e.g., Sharma et al., 2010; Smyth et al., 2007; Poole and Smyth, 2005; Smyth and Poole, 2004). A model is a set of rules defined a priori by an expert, based on scientific literature, making use of the entities and properties defined in the ontology. These models aim

130 to represent expert conceptualized descriptions of a given phenomenon or entity (e.g. landslide susceptibility). The properties used in a model description are concepts stored in the ontology, along with frequency terms (e.g. soil slide – has slope – moderately steep – always). Frequency terms used in this study are: "always", "usually", "sometimes", "rarely" and "never". These terms were chosen as they express experience-based judgements that geoscience practitioners may use in field assessments. The term "never" allows the system to explicitly deal with negation (e.g. soil slide - has surficial material - bedrock - never).

135 The properties and the frequency terms are encoded in semantic triple format (W3C Working Group, 2014) and the resulting model is a semantic network. Semantic networks are a graph representation of knowledge where nodes are concepts and edges are the semantic relation between concepts (Shapiro, 1992); see Figure 2 for example. Real-world areas on the ground (map units – more generally referred to as "instances") are also described by semantic networks using the same properties stored in the ontology, but triples are accompanied by Boolean qualifiers to represent presence or absence of a specific property (e.g.

140 polygon – has slope – steep – present). Comparisons, referred to as matches, between instances and models is possible because models and instances all use the same structured terminology, as controlled by the ontology.

 Similarity scores are awarded based on the type of match between instance and model properties, the semantic distance in the taxonomy of compared property values and the model property frequency term (Figure 2). Match types include, exact, a kind Of (AKO) exact, and possible. An exact match indicates that the property value term used in the model is present in

145 the instance ('a' in Figure 2), in which case full score is awarded for this component of the compared semantic networks. An AKO exact match indicates that the property value term found in the instance is a kind of the property value term found in the model ('b' in Figure 2), in which case a full score is also awarded. A "Possible" match occurs when the property value term in the instance is broader than the property value term in the model, based on the defined taxonomies, in which case the score is divided by the semantic distance between the two terms. For example, 'forest' is a more specific type of 'forest and semi

150 natural areas' ('c' in Figure 2) and results in the score being divided by two. The score is lower because the instance is only possibly the kind of value that the model is looking for.

 In this study, an exact match or an AKO exact match of a property with frequency "always" scores 10,000, "usually" scores 9000, "sometimes" scores 1000, "rarely" scores "100" and "never" scores -10,000; unmatched attributes are awarded -10 points. These scores are an arbitrary representation of degree of surprise that uses order of magnitude numbers to distinguish

155 qualitative measures. For an extensive review of the probabilistic comparison method, see Smyth and Poole (2004) Poole and Smyth (2005), Smyth et al. (2007) and Sharma et al. (2010). This approach has been applied in economic geology to generate mineral deposit exploration targets (Smyth et al., 2007), and in geohazard mapping to produce landslide susceptibility maps (Jackson Jr et al., 2008).

### 2.2.1 Landslide models

160 This paper presents an AI expert-based landslide susceptibility map for three different landslide types: debris flows, slides in soil, and slides in rock for the Veneto region of Italy. These three landslide types are conceptualizations of landslide models defined using knowledge recorded in the scientific literature. These landslide models are intended to be proof-of-concept of models that can be used in the semantic approach proposed in this paper. In particular, some of the properties used in the

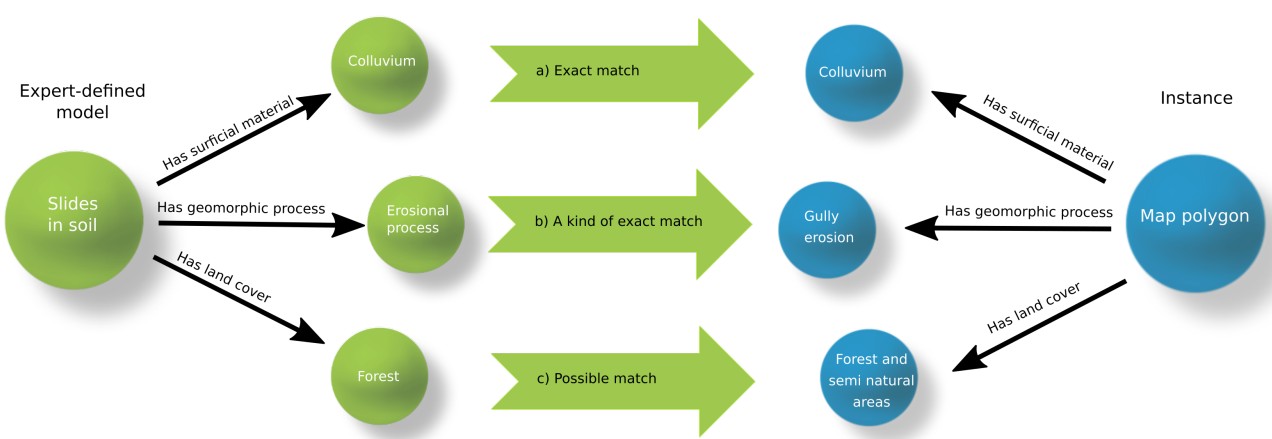

**Figure 2.** Graphical representation of the matching process between expert-defined models and map polygon instances. a) is an example of an exact match between the property value "colluvium"; b) is an example of a kind of (AKO) exact match, because "gully erosion" is a more specific kind of "erosional process". The model is looking for an "erosional process" and found a "gully erosion"; c) is an example of a possible exact match because "forest and semi natural areas" is a broader concept of "forest". The model is looking for "forest" but we do not know whether the instance is a "forest". We only know that the instance is "forest and semi natural areas". The vocabulary and the hierarchy are controlled by the ontology. Note that frequency terms for model properties are not shown in this figure.

models are drafted from literature analysis of logging-related landslides in British Columbia, Canada (Jackson Jr, 2019).
Here we briefly summarize the models; detailed explanations of each property-value-frequency combination are provided in
Appendix C.

The 'Debris Flow' model describes the channels that may generate a debris flow. Debris flows are flow-like landslides
generated when saturated sediments move down a steep channel. They can be originated when a slide in soil intersects a
flowing body of water, or when saturated bed sediments are mobilized and begin flowing downstream. Debris flows are usually
triggered by intense and persistent rainfall (Hungr et al., 2014). To visualize the 'Debris Flow' see the table in Appendix C or
navigate to https://italy.minervageo.com/debris-flow-model/.

The 'Slides in Rock' model describes slopes that may generate slides in rock. Slides in Rock form when steep rock slopes
and cliffs fail under the influence of gravity, and are commonly triggered by intense rainfall or earthquakes. Slides in rock are
usually very fast, and the failure can occur along planar, curved, and/or multiple surfaces. This model represents the collective
class of landslides that have as material "rock" and movement type "slide", including rotational, planar, compound, wedge
and irregular slides in rock (Hungr et al., 2014). Given the regional scale of this study, we do not have the data resolution
to determine the possible failure plane geometry. For example, we cannot identify slopes more susceptible to planar rock

slides rather than rotational rock slides. To visualize the 'Slides in Rock' model see the table in Appendix C or navigate to https://italy.minervageo.com/the-roberti-slides-in-rock-model/

The 'Slides in Soil' model describes slopes that may generate slides in soil. Slides in soil are downslope movements of soil under the influence of gravity, commonly triggered by intense rainfall or earthquakes. They can be slow or fast, and the failure can occur along one or many planar or curved surfaces (Hungr et al., 2014). With Slides in Soil, we refer to the collective class representing all landslides that have as material "soil" and movement type "slide", including rotational, planar, and compound, clay, silt, sand, gravel, debris slides. Given the regional scale of this study, we do not have the data resolution to determine the

possible failure plane geometry and the specific kind of soil that is involved in the failure. To visualize the 'Slides in Soil' see the table in Appendix C or navigate to https://italy.minervageo.com/slides-in-soil/

In the presence of higher resolution information such as rock bedding orientation or shear geometry and stratigraphy in soil masses, specific kinds of rock slides (e.g. planar vs rotational) or different kinds of slides in soil (e. g. clay compound slide vs clay planar slide) susceptibility may be mapped.

### 2.2.2 Map polygon instances

The definition of the mapping unit is a critical step in any landslide susceptibility mapping application and there are many different approaches to subdividing the area of interest to identify areas susceptible to slides in soil or rock (see review by Guzzetti et al., 1999). For this study, we used slope units, which are a geomorphic representation of single slopes bounded by drainage and divide lines (Guzzetti et al., 1999), as mapping unit. We used the r.slopeunits software to automate the slope unit

delineation (Alvioli et al., 2016, 2020). We used stream line vector shapefiles provided by the Veneto Regional Government, buffered by a distance of 5 m as mapping units to map debris flow susceptibility. In total, the region of Veneto was subdivided into 93,262 polygons, of which 9,302 are stream buffer polygons and 83,960 are slope-unit polygons.

We used a spatial overlay analysis to aggregate data describing the physical properties of the mapping units (Figure 6). The analysis aggregated the properties from all features that intersect the mapping units. For each property in an input layer, an

200 aggregation type is specified as either: (a) list, whereby all of the intersecting properties are concatenated into the mapping unit (e.g. multiple rock types), or (b) Boolean evaluation, which checks whether or not the mapping unit was intersected by a specific input feature (e.g. a fault).

The properties describing each mapping unit polygon were converted into semantic networks, one network for each polygon. This conversion allows for semantic reasoning to compare and rank, based on similarity, the mapping units (hereon instances)

against the expert-defined landslide models to evaluate landslide susceptibility.

### 2.2.3 Matching, susceptibility and runout

The similarity score between a given model and instance is used as a proxy of landslide susceptibility. A high similarity score between an instance and a landslide susceptibility model signals a high susceptibility to that type of landslide. We deliver the similarity score between models and instances as susceptibility on the output maps.

**Table 1.** R.avaflow parameters for slides in soil, slides in rock and debris flows runout calculations

| Variables (unit) | Slides in Soil | Slides in Rock | Debris Flow |
|---|---|---|---|
| Solid fraction (%) | 60 | 70 | 60 |
| Fluid fraction (%) | 40 | 30 | 40 |
| Solid fraction internal friction angle (degree) | 18 | 18 | 5 |
| Solid fraction basal friction angle (degree) | 10 | 10 | 4 |
| Fluid fraction internal friction angle (degree) | 0 | 0 | 0 |
| Fluid fraction basal friction angle (degree) | 0 | 0 | 0 |
| Solid fraction viscosity ($m^2$ s-1) | 30 | 30 | 5 |
| Fluid fraction viscosity ($m^2$ s-1) | 3 | 3 | 3 |

After the susceptibility assessment, a first-order estimate of hazard is provided by calculating the likely extent of landslide runout for the most susceptible (99.9 [th] percentile score, i. e. top one in a thousand) instances for each model. Various physical methods have been developed to calculate potential landslide runout, given the physical properties of the material and the topography (see review by McDougall, 2016). To compute the potential runout extents, we applied the r.avaflow code (Mergili et al., 2017) which is an open source software package implementing the two-phase debris flow model developed by Pudasaini (2012). Physical model parameters for 'Slides in rock' are inferred from the back-calculations of the recent Mt. Joffre landslide, in British Columbia, Canada (Friele et al., 2020), 'Slides in soil' and 'Debris flow' parameters use the default r.avaflow parameters for those landslide types (Table 1).

Various landslide size classes were simulated for each map instance, ranging from class 4 to class 6 (Jakob, 2005). Classes 4 to 6 were chosen to provide a preliminary hazard assessment, where class 4 event may have an approximate return interval of hundreds of years and class 6 are very unlikely and extreme events with return intervals on the order of thousands of years (Jakob, 2005).

### 2.2.4 Web map

This study's landslide susceptibility maps and hypothetical landslide runouts for slides in soil, slides in rock and debris flows are delivered as an interactive web map based on OpenLayers (MetaCarta, 2005) and React (Facebook, 2013). Input layers are hosted through a Geoserver (The Open Planning Project, 2001) with a PostGIS (Refraction Researtch, 2001) backend database. INSPIRE-aligned layers are hosted on Hale Connect (WeTransform, 2014), a platform used to host and serve INSPIRE-compliant data.

# 3 Results

## 3.1 INSPIRE Natural Risk Zones extension

To develop an INSPIRE-compliant AI application to map landslide susceptibility, we needed to extend the INSPIRE Risk Zones theme to include the concept of landslide susceptibility and the specific code list dealing with landslide terminology. The INSPIRE extensions developed in this project are documented and stored in the Minerva Re3gistry (Minerva Intelligence, 2019a), a version 1.3.1 of the INSPIRE registry based on the Re3gistry software (ISA, 2016). The registry service is packaged within a collection of Docker (Hykes, 2013) containers and hosted on a local server.

The Natural Risk Zone core (NZ-core) schema extension, which includes the Natural Risk Zone Susceptibility feature type was based on SafeLand recommendations (SafeLand, 2011). The Natural Hazard Classification code list was extended (Minerva Intelligence, 2019b) to include a classification of various landslide types using the updated Varnes landslide classification (Hungr et al., 2014), which is a landslide classification widely adopted within the scientific community, and a new code list of landslide size classes (Minerva Intelligence, 2019c) based on Jakob (2005). The landslide size code list contains ten landslide

size classes based on landslide volume and descriptions of approximate damage potential.

### 3.1.1 Code list extension

The Natural Hazard Classification code list extension for landslides considers material type and failure movement, splitting the tree, first on type of movement, and then on type of material, following Hungr et al. (2014) (Figure 3). Other properties, such as: water content, depth of failure, rate of movement, loading state, channelized state, and failure plane geometry (see Appendix

B) are used to describe the individual landslide types, as the unique combination of these properties allows for unambiguous classification in an Aristotelian taxonomy. We used these properties because, even if not shown in the final taxonomic tree, they are explicitly applied in the wordy description of landslide types by Hungr et al. (2014).

The formal extension registration process via the INSPIRE Registry software does not enable the representation of such multi-hierarchical classifications. Because of this we had to work with a single tree hierarchy, and consequently chose to first

divide the classes on type of failure followed by a division based on the type of movement (Figure 3).

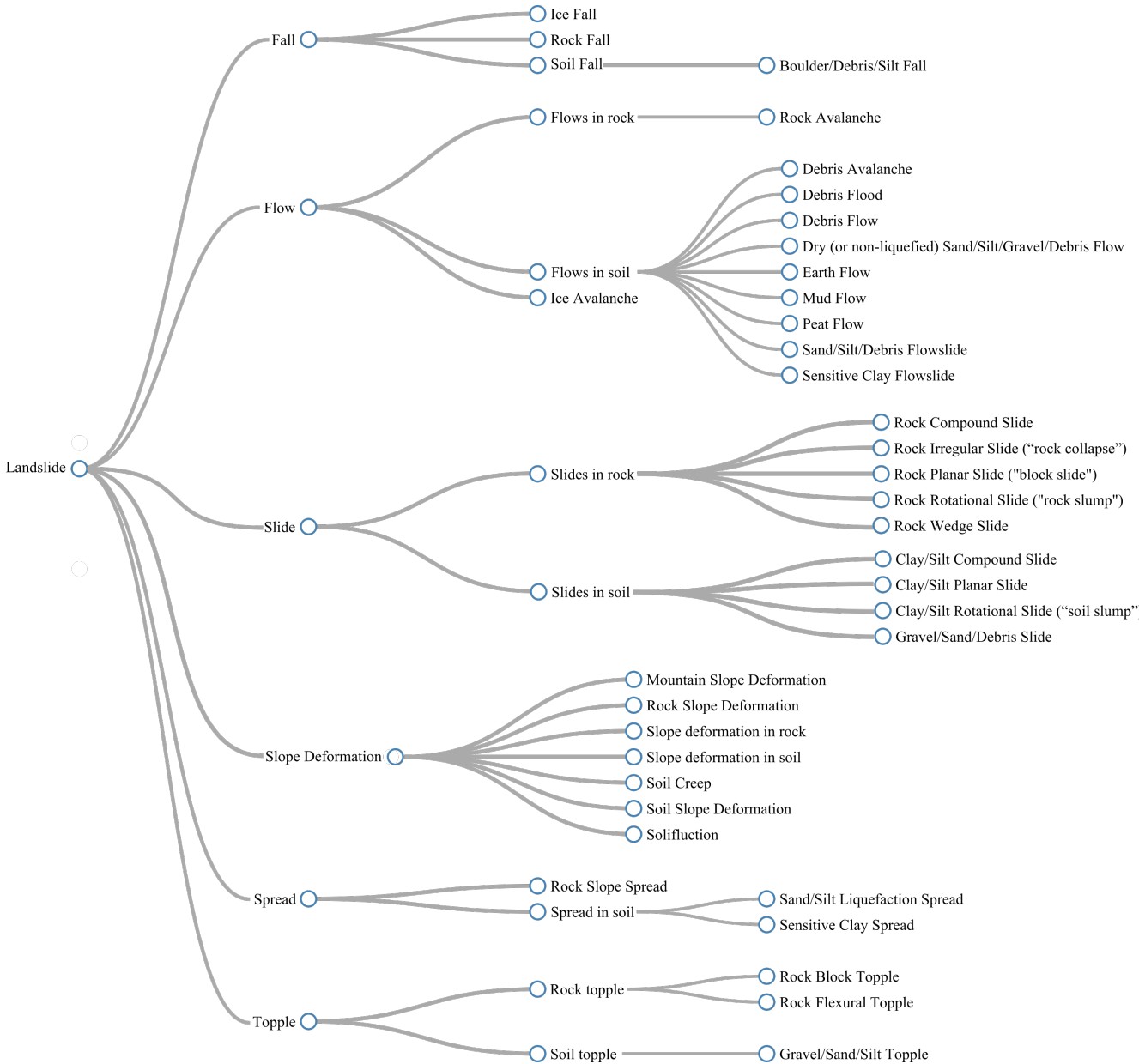

**Figure 3.** Natural Hazard Category code list extension for landslides

### 3.1.2 Schema extension: susceptibility

The INSPIRE Natural Risk Zone schema includes hazard and risk feature types, but the concept of susceptibility as a feature type is missing. To overcome this problem, we extended the INSPIRE Natural Risk Zone core XML schema, adding a Natural Risk Zone Susceptibility schema (Minerva Intelligence, 2019e). The Natural Risk Zone Susceptibility schema includes Abstract Susceptibility Area and Susceptibility Area feature types (Figure 4). The Susceptibility Area feature type is modelled following the structure of the Hazard Area and Risk Zone feature types in the INSPIRE Natural Risk Zone core schema. Susceptibility Area has three elements: Geometry, Influencing Factor and Relative Spatial Likelihood of Occurrence (Figure 4). Geometry, as with all INSPIRE vector datasets, is the geometric representation of the extent of the feature on the Earth Surface as a spatial feature. Influencing factors are defined as the intrinsic, preparatory variables which make an area susceptible to a hazard (SafeLand, 2011). Influencing factors are unbounded in multiplicity (i.e. can be as many as needed) and can be defined qualitatively or quantitatively. Qualitative influencing factors are expressed as a string, while quantitative influencing factors are expressed as GML:MeasureType (Figure 4). Whether defined quantitatively or qualitatively, the influencing factor can also define a DataSetType attribute, such as slope or air quality. Influencing factors are used in the calculation of Relative Spatial Likelihood of Occurrence, which is an element that can be quantitatively or qualitatively defined (Figure 4). The relative spatial likelihood of occurrence refers to values that represent the spatial probability of occurrence of a specific hazard type, given the influencing factors present in the area (SafeLand, 2011). The Influencing Factor element allows end users of Susceptibility Area datasets to understand which known conditions of the specific area led to the resultant susceptibility.

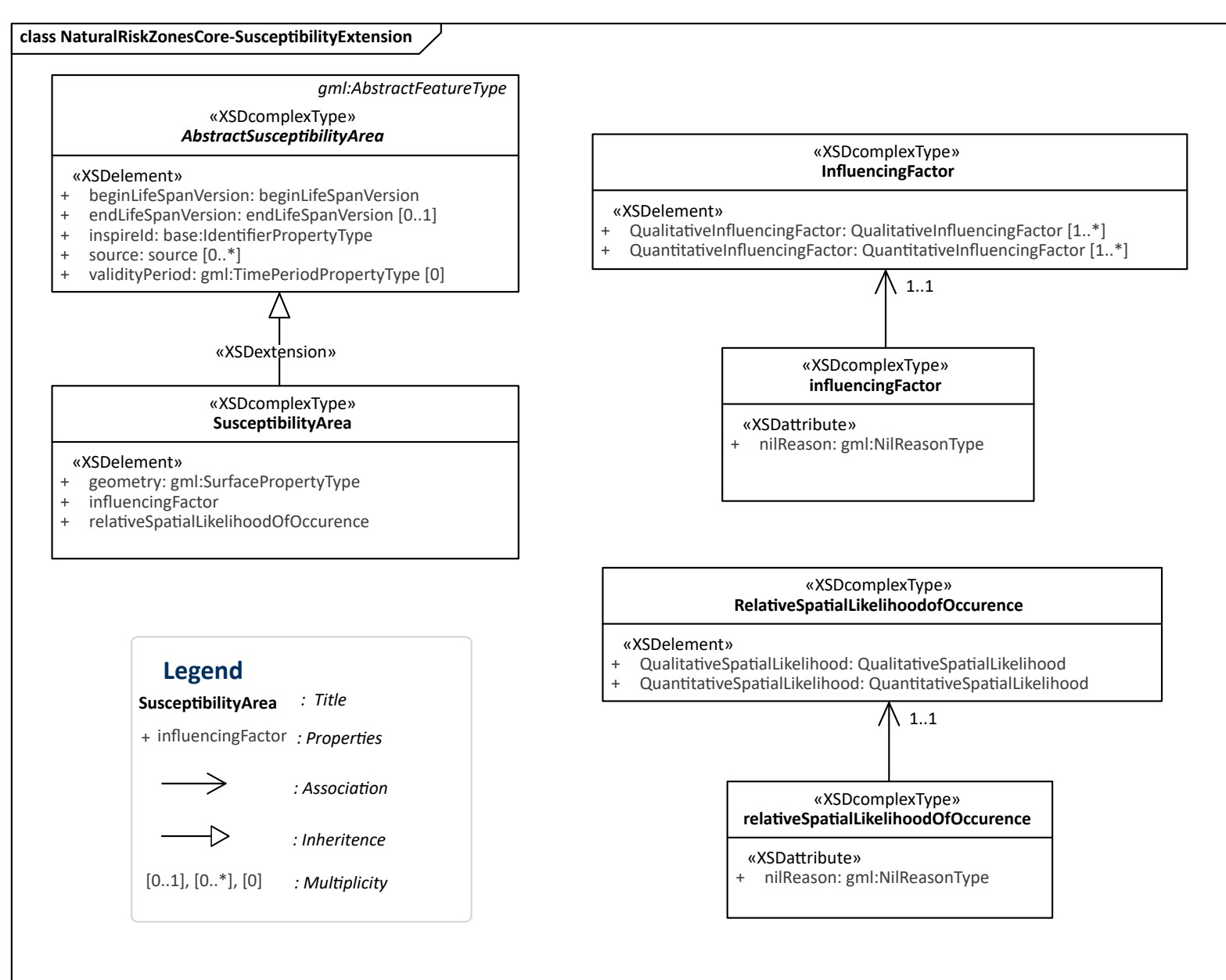

**Figure 4. UML diagram showing Natural Risk Zone Susceptibility schema extension of the Natural Risk Zone Core Schema**

## 3.2 Landslide susceptibility mapping in Veneto

### 3.2.1 Input data

For this study, we used open access datasets from the Veneto Region Geoportal and other sources (Table 2 and 3). Aligning all input datasets was beyond the scope of this project. We did, however, want to show the value of INSPIRE-aligned data and therefore aligned stream network, CORINE land cover, bedrock geology, and the Italian Landslide Inventory (IFFI) (Table 2) to INSPIRE using Hale Studio (WeTransform, 2008). Figure 5 shows how different tools in Hale Studio are used to align properties from the source dataset to the target dataset. For example, the field "eta" –"Age" in Italian, of the original Veneto dataset, was directly mapped to four different INSPIRE fields: the olderNamedAge.href and title and the youngerNamedAge.href and
title. Note that olderNamedAge.href youngerNamedAge.href are hyperlinks to the code list value id and the title is the actual code list term from the GeochronologicEraValue code list. This alignment is done with many classification methods, including: Groovy Scripts, formatted strings and assign-alignment tools. For further explanation on term alignments, refer to the documentation of Hale Studio (WeTransform, 2008). Datasets used that were not compliant with INSPIRE include: lakes,
watersheds, permafrost, fire, slope angle, faults, soil, roads and railways (Table 3).

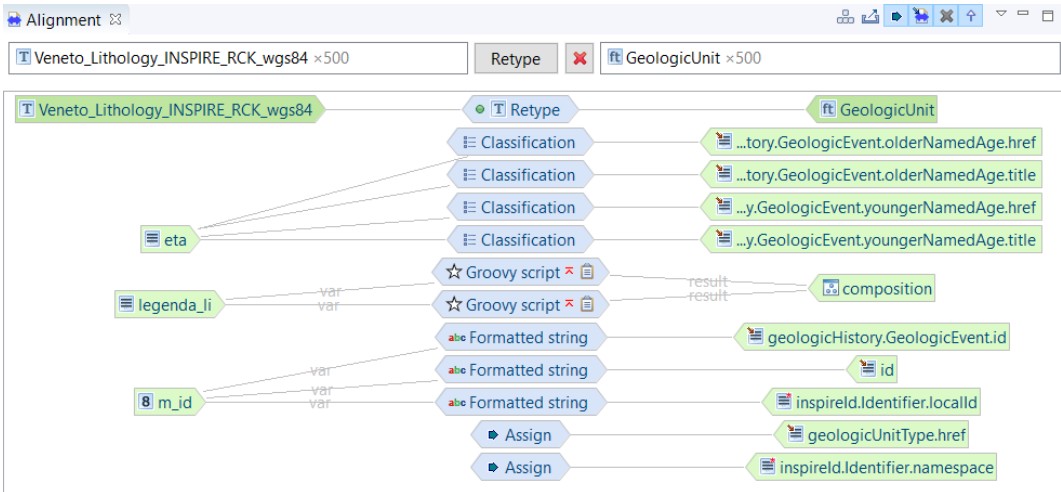

**Figure 5.** INSPIRE alignment visualization within Hale Studio showing the alignment from the Veneto bedrock geology Veneto Lithology "INSPIRE RCK wgs84" shapefile source (left side) to the target "Geologic Unit" feature type within the INSPIRE Geology Schema (right side).

**Table 2.** INSPIRE-compliant layers

| Layer | Description | Source URL (last access: January 2020) |
|---|---|---|
| Streams | Hydrographic network in the Veneto region, including streams, rivers, and other inland flowing water bodies | https://idt2.regione.veneto.it |
| Land Cover (CORINE) | Land cover units in the Veneto region. The CORINE Land Cover (CLC) classification which includes 44 classes, and was last updated in 2018 | https://land.copernicus.eu/pan-european corine-land-cover |
| Geology | Bedrock lithology in the Veneto region. | http://www.pcn.minambiente.it/mattm/ en/wfs-service/ |
| IFFI Landslide Points and Areas | Landslides that have been identified in the Veneto region as part of the IFFI project. The INSPIRE Natural Hazard Category code list was extended to include the updated Varnes landslide classification (Hungr et al., 2014), and the data were aligned to this standard | http://www.pcn.minambiente.it/mattm/ en/wfs-service/ |

**Table 3.** Layers not compliant with INSPIRE standards

| Layer | Description | Source URL (last access: January 2020) |
| --- | --- | --- |
| Lakes | Lakes in the Veneto region. | https://idt2.regione.veneto.it |
| Watersheds | Watersheds in the Veneto region, derived from a digital elevation model from the TINITALY project made available by the National Institute of Geophysics and Volcanology (INGV). | http://tinitaly.pi.ingv.it/ |
| Permafrost | Permafrost derived from the Global Permafrost Zonation Index Map (Gruber, 2012) | http://www.geo.uzh.ch/microsite/cryodata/ |
| Fires | Location and date of past forest fires in the Veneto region. | https://idt2.regione.veneto.it |
| Slope | The gradient of the slope in the Veneto region, derived from a digital elevation model from the TINITALY project made available by the National Institute of Geophysics and Volcanology. | http://tinitaly.pi.ingv.it/ |
| Faults | Faults in the Veneto region, published as part of the Database of Individual Seismogenic Sources (DISS) provided by the National Institute of Geophysics and Volcanology (INGV). | http://diss.rm.ingv.it/diss/index.php/DISS321 |
| Soils | Soil map of the Veneto region, including information about surficial deposit genesis, material, texture, thickness, geomorphic form and process. | https://idt2.regione.veneto.it |
| Railroads | Railroad network in the Veneto region. | https://idt2.regione.veneto.it |
| Roads | Road network in the Veneto region. | https://idt2.regione.veneto.it |

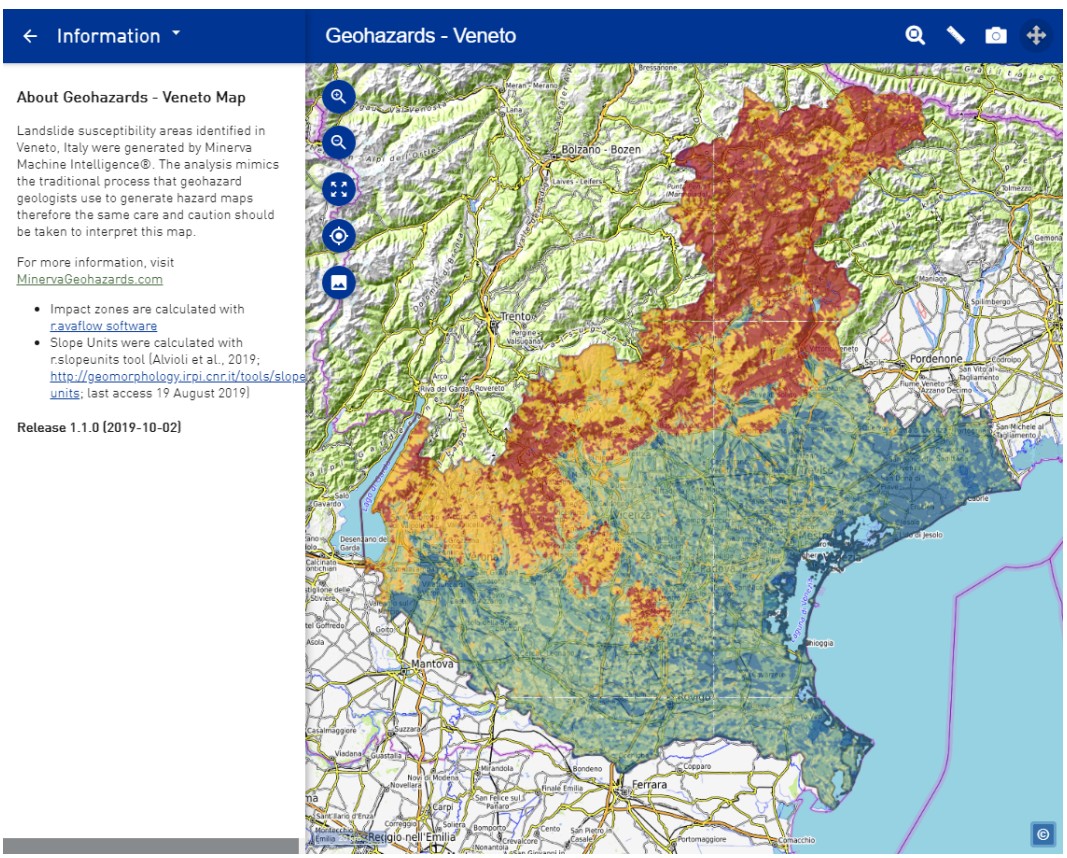

**Figure 6.** Web map interface showing susceptibility to slides in soil in Veneto, Italy. Colours range from blue (0-20 score percentile), to dark green (20-40 score percentile), light green (40-60 score percentile), yellow (60-80 score percentile), red (80-99.9 score percentile) and purple (99.9-100 score percentile). Base map credit: © OpenTopoMap (CC-BY-SA)

### 3.2.2 Webmap

The 83,960 slope units and 9,302 stream buffer instances (Figure 6) are encoded with the available data, then transformed from vector files into semantic network format. Then, each polygon was matched against the expert-based slides in soil, slides in rock and debris flow models and colour coded on matching score percentile to portray landslide susceptibility (Figure 6). The left-side panel of the webmap shows the landslide model layers, the reference layers, and different base maps (Figure 7). By clicking on a polygon (instance), a popup window opens (Figure 7): this window contains the name and hyperlink to the INSPIRE registry code list definition of the landslide type investigated, the attributes that are present in the mapping unit (e.g. bedrock lithology, erosional process, etc.), the instance percentile rank and total match score, the hyperlink to the comparison of the instance against other landslide models (e.g the slides in rock model), and (only for the 99.9[th] percentile score, top one in

**Table 4.** Simplified match report table showing instance 117309 compared to slides in soil mode. The match report is accessible online by clicking https://spot.italy.minervageohazards.com/match_results?if_id=34434&t_id=117309

| Model | | | Instance | | | Results | |
|---|---|---|---|---|---|---|---|
| Property | value | freq. | Property | value | freq. | match type | Score |
| has Geomorph Process | ErosionalProcess | always | has Geomorph Process | Gully Erosion | present | AKO match | 10000 |
| has Surficial Material | Colluvium | always | has Surficial Material | Colluvium | present | exact match | 10000 |

**Table 5.** Simplified match report table showing the comment for the model property "has erosional process" matching the instance property "Gully erosion". The full match report is accessible online at https://spot.italy.minervageohazards.com/match_results?if_id=34434&t_id=117309

| Model | Instnce | Comment | Original value |
|---|---|---|---|
| Erosional process - Always | Gully erosion - Present | Active erosional processes are possible indicators of landslide activity, as landslides occur where landslides have occurred before. | Rock fall, gully erosion, erosional process, karst. |

one thousand) buttons to turn on the display of landslide runout for different landslide classes, and the hyperlink to the match report.

The match report is a detailed table showing the results from the model-instance semantic matching, ensuring the explainability of the results. Each line corresponds to a property-value-frequency term (e.g has slope – moderately steep – always) comparison between the model and the instance, how they match (with a hyperlink to textual explanation on how the score was awarded), the numerical score value, (see Table 4 for example) a textual explanation on why that attribute was chosen, and the original data value (Table 5). An "advice" button opening a textual advice expressing which of the instance unmatched attributes may change the score is available. This advice is a sort of data-advice: it invites the user to check in the field or in some other databases if, for example, a fault is present in that specific instance.

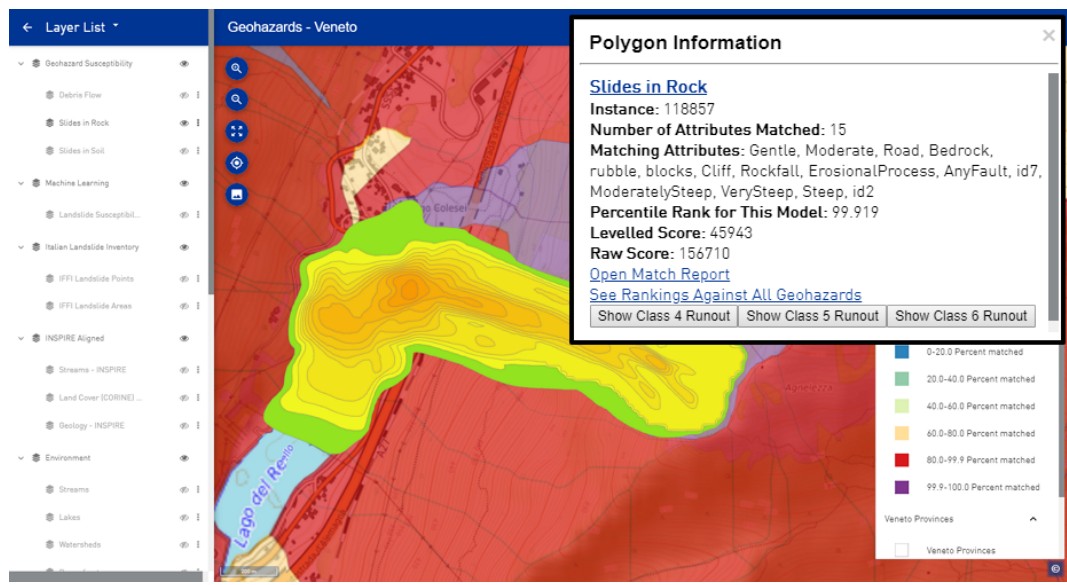

**Figure 7.** Screen capture showing left side panel with the layer list, information popup window, map legend, and the landslide runout. Base map credit: © OpenTopoMap (CC-BY-SA)

## 4 Discussion

### 4.1 INSPIRE as a framework for Explainable AI

Across society, the use of numerous complex and non-standardized earth science taxonomies results in interoperability limitations, which hinder the widespread implementation of explainable AI solutions to natural hazard-related problems. This is evident in the landslide domain, where data layers for landslide susceptibility analysis, ranging from landslide databases (Van Den Eeckhaut et al., 2013) to geomorphology maps, vary across regions and countries. Consequently, despite the wealth of scientific literature on landslides in general, and landslide susceptibility in particular (Reichenbach et al., 2018), broad-scale operational landslide hazard management systems are scarce (Guzzetti et al., 2020), resulting in significant human and economic losses (Froude and Petley, 2018).

INSPIRE partially addresses this problem by providing standardised data structures for data-hosting and standard terminology to use within those structures. This study illustrates that, once INSPIRE-compliant, European data can be subjected to analytical methods that can be applied for practical application to multiple other equivalent INSPIRE-compliant data sets. For example, the same landslide-focused ontology that uses terminology and knowledge models based on INSPIRE code lists used in this project has been applied in South-Western British Columbia, Canada (Minerva Intelligence, 2019f).

By maintaining carefully curated standards, INSPIRE can play a critical role in AI applications that seek to be "explainable" (Gilpin et al., 2019). Its code lists can be mapped into ontology properties, enabling machines to make inferences of semantic and hierarchic relations based on data. The explainability in the application presented in this study is provided in the form of a comprehensive match report, which can be opened via an information popup for each instance. The match report provides the user with complete access to the logic that drives the AI reasoning engine, allowing interrogation of the results displayed on the map. By embedding explanations in a user-friendly interface, ontologically-based AI can improve the understanding of complex geospatial problems by decision-makers, insurance companies and the general public.

Public and private organisations, within and outside the European Union, can significantly enhance the value of the data they collect and publish by using INSPIRE-compliant standards not only in natural hazard mapping but also in other domains. A comparative study of regional SDI in the context of INSPIRE implementation (Craglia and Campagna, 2010) showed that inefficient data access and use at the European level results in economic losses in the 100-200 Million Euro annual range. The same study, shows that the regional SDI of Lombardia, Italy, allowed 3 Million Euro per year savings to companies working in Environmental Impact Assessments (EIA), and Strategic Environmental Assessments (SEA). Savings in the same order of magnitude can be expected by adopting INSPIRE standards in the geological hazard assessment domain.

### 4.2 INSPIRE extension and limitations

INSPIRE-compliant datasets are still rare across European countries in general, and in Italy in particular (Cetl et al., 2017; Mijić and Bartha, 2018; Cho and Crompvoets, 2019). Consequently, we were unable to identify a jurisdiction in Europe with INSPIRE-compliant datasets for all the inputs necessary for this study. Therefore, instead of using already-compliant data, a region optimal for demonstrating the inter-relationship between INSPIRE and explainable AI was chosen, and some of the data

for that region was transformed into INSPIRE compliance. In doing so, the study provides both a case study of dealing with non-INSPIRE-compliant data, and an illustration of the rewards achievable by bringing a coherent set of data into INSPIRE compliance.

The code lists and application schemas in the INSPIRE Natural Risk Zone theme lacked the level of detail necessary for this application. This is understandable, as given the broad scope of the directive, schemas lack the necessary granularity for specific applications. INSPIRE is intended to be used as an overarching umbrella under which domain-specific applications can find their place by extending it where necessary. The Natural Risk Zone theme (Tomas et al., 2015) and the extension presented in this work is an example of using this extension facility. Within the Natural Risk Zone theme, the Natural Hazard Category Value

code list includes geological/hydrological hazards, including 'flood' and 'landslide', but the different subclasses of floods and landslides are not specified. For this kind of landslide susceptibility assessment, the clear definition of landslide types, landslide size classes, and susceptibility was fundamental. For example, a debris flow, which moves rapidly (meters per second), and an earth flow, which may move slowly (meters per year) present entirely different hazards; they can both destroy property but it is unlikely for an earth flow to result in fatalities while the opposite can be said of debris flows (Hungr et al., 2014). The definition

of landslide sizes is also important: a size class 1 debris flow has a smaller impact area than a size class 6 event, but, by having a higher frequency, it may result in greater losses (Jakob, 2005).

From a data structure perspective, INSPIRE code lists cannot currently host multi-hierarchical taxonomies. This limits the nature of reasoning that can be brought to bear on them. We understand the technical difficulties in handling multi-hierarchical taxonomies, but hope that future versions of the Registry software will be able to handle these complex knowledge representa-

tions.

The INSPIRE Natural Risk Zone theme also lacks the definition of susceptibility as a concept and feature type. The term susceptibility is not implemented as a feature type because for most hazards (e. g floods and earthquakes) the concept is embedded within the concept of hazard likelihood (Tomas et al., 2015). This does not apply in the landslide domain where susceptibility and hazard are distinct concepts (e.g. Van Den Eeckhaut and Hervás, 2012). In this study, we implemented the

susceptibility feature type. Although we applied this feature type in the landslide domain, it will be useful for other natural hazard applications, when the spatial likelihood of hazard occurrence must be expressed separately from the general concept of hazard likelihood.

The extensibility of INSPIRE allows for domain-specific applications, like the approach presented in this paper, to fit within the INSPIRE framework. However, problems may also arise from the fact that INSPIRE is extensible. Extensibility allows

greater precision in terminology and schema for a specific application but this allows different public and private institutions to implement separate, and eventually, incompatible extensions. For example, another landslide classification may be implemented by another institution: this implementation may not be interoperable with the one presented in this study, but will have the same INSPIRE compliance, leading to two conflicting standards. Much work remains at the level of thematic clusters to implement as many standardized vocabularies and schemas as possible. Our extension is open and free, and we hope that other

entities will adopt it for other landslide applications.

### 4.3 Ontological probabilistic matching for landslide susceptibility mapping

The semantic AI system applied in this study aimed to replicate the reasoning with uncertainties typical of geological assessments, applying the terminology that geological and geotechnical professionals use in their daily practice (Smyth et al., 2007). Since they are based on expert-defined models, the landslide susceptibility maps produced in this study are comparable to qualitative heuristic assessments (SafeLand, 2011). The choice of using a qualitative method for landslide susceptibility assessment is in contrast with recent recommendations for the application of quantitative methods (Corominas et al., 2014). However, in current professional geological assessments and geomorphological mapping applications, expert judgment is still widely applied (e.g., Association of Professional Engineers and Geoscientists of British Columbia, 2010; Guzzetti et al., 2012), and quantitative (statistically and physically-based) methods rely on data that are not always available or of unknown quality. For example, landslide databases necessary for statistically-based susceptibility mapping are often incomplete, inaccurate, and geographically-limited (Guzzetti et al., 2012). Further, the geotechnical parameters necessary for running physical models are usually approximated to carry out regional-scale studies (e.g., Mergili et al., 2014).

The semantic AI system applied in this study can be used in cases of data scarcity, and if coupled with numerical methods, can improve the explainability of predictions. For example, by embedding the ontology concepts related to statistical parameters (e.g. receiving operating curves, confidence intervals) or physical parameters (e. g. friction angles, viscosity), it will be possible for the numerical outputs of quantitative methods to be explained in natural language, helping to reduce the gap between scientists and decision-makers (Newman et al., 2017).

The main goal of this paper is not to present the semantic matching approach, but to show an example on how to modify INSPIRE to make it possible to use it for landslide-specific applications. By suggesting these landslide-specific schema and code list extensions, we set the ground for INSPIRE-compliant landslide susceptibility studies. Other organizations can build on top of these extensions and future landslide susceptibility applications can be compared as they formally refer to the same data structure and semantics. Note that we do not force any specific data and modeling variable selection, nor modeling approach for landslide susceptibility/hazard/risk method. Such an effort is beyond the scope to this paper and, to some extent, already addressed by the SafeLand project (e.g., SafeLand, 2011) rather, we provide the data structure and semantics to store and share whichever method has been chosen by the modeler. For example, data selection for calculation of landslide susceptibility is encompassed in the schema structure under "Influencing Factor" which are "unbounded in multiplicity and can be defined qualitatively or quantitatively", leaving broad range of possibilities to the modeler. Regarding the data quality, it is discussed in the Natural Risk Zone schema and they refer to ISO standards (INSPIRE Thematic Working Group Natural Risk Zones, 2013). However, we recognize that specific code list (semantics) dealing with data quality and model uncertainty are missing. We hope that the INSPIRE thematic group will address this point.

## 5 Conclusions

This study presents an AI method, based on semantic network comparison, to produce landslide susceptibility maps using an ontology and standardized taxonomies within the framework provided by the INSPIRE Natural Risk Zone theme. This

method does not need an accurate landslides inventory to make predictions, as it uses qualitative probabilistic reasoning that incorporates expert knowledge. We produced susceptibility maps for debris flow, slides in soil and slides in rock for the province of Veneto, Italy. To produce the maps for specific landslide types, we extended the Natural Risk Zone theme to encompass both the concept of susceptibility and the different types of landslides. In particular, we registered a landslide classification extension of the Natural Hazard Category code list, a landslide size class code list, and Susceptibility Area and Abstract Susceptibility Area feature types schema extensions. After defining the extension, we aligned key input layers (geology, streams, and land cover) to INSPIRE and, by using an ontologically-grounded probabilistic matching algorithm, we produced the landslide susceptibility layers. The processing outputs were mapped to the Natural Risk Zone Susceptibility schema extension. Then, potential impact zones of landslides for multiple landslide-size classes were physically modelled for a subset of the instances with the highest susceptibility scores. Finally, the results were presented in a user-friendly interface, embedding plain language explanations on how the score was assign and advises on how to improve the matching.

We have demonstrated the value of INSPIRE-compliance by showing how it enhances information and knowledge interoperability, and allows for explainability in AI applications by standardized interrogation of their inputs and outputs. Ontologies provide the formal structure for INSPIRE code lists to run algorithms similar to that applied here. The maps can explain the scientific results that they portray, and consequently improve the understanding of complex geospatial problems not only by domain experts but also by decision-makers and other non-specialized interested parties.

This study also illustrates that, in their current state of development, the INSPIRE standards are not sufficiently expressive to support complex landslide susceptibility mapping. We provided an example of how INSPIRE's extension capabilities may be implemented to add the required expressivity. This extension framework ensures, through its Re3gistry register, that the expressivity extensions are documented and available to all interested parties for re-use. In so doing, it sets the context for the ongoing refinement of standards by the INSPIRE thematic committees.

## 6  Data availability

- The web application is available at: https://map.italy.minervageohazards.com/

- The schema extension is available at: https://github.com/minervaintelligence/INSPIRE-NZ-Susceptibility

- The code list extension is available at: http://minerva.codes/registry

- Data from the Italian National geoportal is available under "Attribution-NonCommercial-ShareAlike 3.0 Italy (CC BY-NC-SA 3.0 IT)" License, https://creativecommons.org/licenses/by-nc-sa/3.0/it/deed.en

- Data from the Veneto Geoportal are available under the "Italian Open Data License 2.0", https://www.dati.gov.it/content/italian-open-data-license-v20

- CORINE land cover data is available under EEA standard re-use policy: re-use of content on the EEA website for commercial or non-commercial purposes is permitted free of charge, provided that the source is acknowledged (http://www.eea.europa.eu/legal/copyright)

- Tinitaly DEM is available upon request by sending an email to simone.tarquini@ingv.it with the subject of TINITALY DEM. Terms and Conditions of Use: Data is provided for research purposes only. Data is provided solely to the person named on this application form and should not be given to third parties. Third parties who might need access to the same dataset are required to fill their own application forms http://tinitaly.pi.ingv.it/ Data from INGV is available under "Creative Commons Attribution-ShareAlike 4.0 International (CC BY-SA 4.0)" license http://creativecommons.org/licenses/by-sa/4.0/

- The permafrost data is available under "Attribution 3.0 Unported (CC BY 3.0)" licence. http://www.geo.uzh.ch/microsite/cryodata/.

# B   Appendix A - Dictionary of Terms

| Term | Description |
| --- | --- |
| Code list | A dataset specifying terms for populating INSPIRE properties that require controlled vocabulary |
| CLC | CORINE land Cover, a classification system for land cover based on vegetation and land use |
| Feature type | A data type representing a thematic entity in a domain of interest, typically with some geospatial location specified by vector based spatial data |
| IFFI | Italian Landslide Inventory |
| Instance | A data item that represents an individual, specific real-world entity, for this application an instance is a spatial feature either a slope unit polygon or a stream buffer polygon. |
| Model | A conceptualization of the entities, properties and relationships in some domain of interest, in this case, landslides. Three landslide models were used in this project; debris flow, slides in soil and slides in rock. |
| Ontology | A formal representation of a conceptualization of the entities, properties, relationships, and rules describing the relation between the entities in some domain of interest. |
| Semantic Network | A graph network of arcs and nodes that represent concepts in a domain of interest. |
| Schema | A representation of a data model, describes the structure of a data theme |
| Slope unit | A map unit polygon that is derived from the digital elevation model, defined by hydrologic drainage and divide lines |
| Taxonomy | Hierarchical classification scheme based on shared characteristics between entities |
| Triple | A semantic triple is a subject-object-predicate expression that asserts a fact, and it is the basic unit of a semantic network. |

# D    Appendix B - Properties used for the landslide classification

| Property | Property definition | Property value | Property value definition |
|---|---|---|---|
| Type of movement | Landslide movement types (Hungr et al., 2014) | Fall | A fall starts with the detachment of soil or rock from a steep slope along a surface on which little or no shear displacement takes place. The material then descends largely through the air by falling, saltation or rolling (Cruden and Couture, 2011) |
| | | Topple | A topple is the forward rotation of material about a point or axis below the centre of gravity of the displaced mass. (Cruden and Couture, 2011) |
| | | Slide | A slide is a downslope movement occurring dominantly on surfaces of rupture or relatively thin zones of intense shear strain (Cruden and Couture, 2011) |
| | | Spread | Spread is an extension of mass combined with a general subsidence of a upper fractured mass of material into softer underlying material. (Cruden and Couture, 2011) |
| | | Flow | A flow is a spatially continuous movement in which surfaces of shear are short-lived, closely spaced and not usually preserved (Cruden and Couture, 2011). |
| | | Slope deformation | Slow, sometime unmeasurable, deformation of slopes (Hungr et al., 2014) |
| Material | Landslide-forming material types (Hungr et al., 2014) | Ice | Glacier ice or other solid water on steep slopes (Hungr et al., 2014) |
| | | Rock | Intrusive, volcanic, metamorphic, strong sedimentary, (carbonatic or arenaceous) and weak sedimentary (argillaceous) (Hungr et al., 2014) |

| | | | |
|---|---|---|---|
| | Strong | | Rock broken with hammer (Hungr et al., 2014) |
| | Weak | | Rock peeled with knife (Hungr et al., 2014) |
| Soil | | | Residual, colluvial, alluvial, lacustrine, marine, aeolian, glacial, volcanic, organic, random anthropogenic fills, engineered anthropogenic fills, mine tailings, and sanitary waste (Hungr et al., 2014). |
| | Peat | | Organic material (Hungr et al., 2014). |
| | Debris | | Low plasticity, unsorted and mixed material (Hungr et al., 2014). |
| | Silt, sand, gravel, and boulders | | Nonplastic (or very low plasticicty), granular, sorted. Silt particles cannot be seen by eye. (Hungr et al., 2014). |
| | | Partly saturated | GW, GP, and GM unified soil classes (Hungr et al., 2014). |
| | | saturated | SW, SP, and SM unified soil classes (Hungr et al., 2014). |
| | | dry | ML unified soil class (Hungr et al., 2014). |
| | Mud | | Plastic, unsorted, and close to Liquid Limit material. CL, CH, and CM unified soil classes (Hungr et al., 2014). |
| | Clay | | Plastic, can be modeled into standard thread when moist, has dry strength. GC, SC, CL, MH, CH, OL, and OH unified soil classes (Hungr et al., 2014). |
| | | Sensitive | Sensitive or quick clay is a special type of clay prone to sudden strength loss upon disturbance. From a relatively stiff material in the undisturbed condition, an imposed stress can turn such clay into a liquid gel (Geertsema, 2013). |
| | | soft | Easily molded with fingers. Point of geologic pick easily pushed into shaft of handle. Easily penetrated several centimeters by thumb. (Hungr et al., 2014; USDA, 2012). |
| | | stiff | Indented by thumb with great effort. Point of geologic pick can be pushed in up to 1 centimeter. Very difficult to mold with fingers. Just penetrated with hand spade (Hungr et al., 2014; USDA, 2012). |

# E  Appendix C - Landslide models

**Table E1.** Debris flow model https://italy.minervageo.com/debris-flow-model/

| Instance Property-Value-Frequency | Model Definition Source | Comments |
|---|---|---|
| has surficial form -Fan(s)-always | (Goudie, 2014) | Fans are where debris flows deposit. |
| has surficial form -Terrace(s)-usually | (Goudie, 2014) | Terraces are formed by downcutting and lateral erosion of alluvial sediments by streams. Debris flows can generate terraces; hence, terraces can be indicator of debris flow activity. |
| has surficial form -Hummock(s)-always | (Howes and Kenk, 1997) | Hummocky topography may be indicator of landslide debris |
| has water -River/Stream-always | (Howes and Kenk, 1997) | Debris flows occur periodically on established paths, usually gullies and first- or second- order streams |
| has rainfall - Extreme Rainfall- always | (Friele, 2012; Segoni et al., 2018) | Debris flows are triggered by intense rainfall (Segoni et al., 2018). Rainfall threshold for this study are derived from Friele (2012). |
| has rainfall -Severe Rainfall-usually | (Friele, 2012; Segoni et al., 2018) | Debris flows are triggered by intense rainfall (Segoni et al., 2018). Rainfall threshold for this study are derived from Friele (2012). |
| has rainfall -Moderate Rainfall-sometimes | (Friele, 2012; Segoni et al., 2018) | Debris flows are triggered by intense rainfall (Segoni et al., 2018). Rainfall threshold for this study are derived from Friele (2012). |
| has rainfall -Mild Rainfall-rarely | (Friele, 2012; Segoni et al., 2018) | Debris flows are triggered by intense rainfall (Segoni et al., 2018). Rainfall threshold for this study are derived from Friele (2012). |
| has geomorph process - ErosionalProcess-always | (Bovis and Jakob, 1999)) | Streams with active erosional processes are more likely to experience debris flows than streams with less active erosional. |

| | | |
|---|---|---|
| has geomorph process - MassMovement-always | (Guzzetti et al., 2012) | Landslides are more likely to occur on slopes or valleys that have experienced landslides before |
| has been logged within years -5-10 years-always | (Jackson Jr, 2019) | Landslides are extremely likely by 5 to 10 years after tree harvesting. Most of tree roots have died, and new trees are too small to provide anchoring effect with their roots on the slope. |
| has been logged within years -10-20 years-usually | (Jackson Jr, 2019) | Landslides are likely by 10 to 20 years after tree harvesting as new trees are starting to provide anchoring effect with their roots on the slope. |
| has been logged within years -0-5 years-usually | (Jackson Jr, 2019) | Landslides are likely by 0 to 5 years after tree harvesting as the trees are dead but some roots are still providing anchoring effect on the slope. |
| has fire within years -0-2 years-always | (Jackson Jr, 2019) | Debris flows are very likely for 2 years after a wildfire. Water cannot infiltrate, runoff and erosion increase as the soil becomes water repellent and loses cohesion because of the fire heat. |
| has fire within years -3-5 years-usually | (Jackson Jr, 2019) | Debris flows are likely between 3 to 5 years after a wildfire. The water-repellent soil horizon degrades but the roots of dead trees are starting to rot and they do not support the slope with their anchoring effect anymore. |
| has fire within years -5-10 years-always | (Jackson Jr, 2019) | Debris flows are very likely between 5 to 10 years after a wildfire. Roots of dead trees decay, and they are not supporting the soil anymore as for the case of tree harvesting logging. |
| has fire within years - 10-20 years- usually | (Jackson Jr, 2019) | Debris flows are likely between 10 to 20 years after a wildfire. The roots have lost anchoring effect and the new trees are still too small to support the slope. |

| | | |
|---|---|---|
| has transport line -Road Resource-always | (Jackson Jr, 2019) | Logging roads are the greatest aggravating factor for landslide activity as compared to undisturbed slopes. |
| has transport line -Road Resource Demographic-always | (Jackson Jr, 2019) | Logging roads are the greatest aggravating factor for landslide activity as compared to undisturbed slopes. |
| has transport line -Road Unclassified Or Unknown-always | (Jackson Jr, 2019) | The 'Road Unclassified Or Unknown' in this area of BC are mostly old inactive logging roads. This assessment has been done by visual evaluation of the data. Logging roads are the greatest aggravating factor for landslide activity as compared to undisturbed slopes. |
| has bed rock -volcanic igneous rock-always | (Bovis and Jakob, 1999)) | Quaternary volcanic rocks in BC have usually weak geotechnical properties. Basin underlain by these weak rocks are likely to experience frequent and large debris flow events. |
| has fire within years ->20 years-sometimes | (Jackson Jr, 2019) | After 20 year since a wildfire, trees have regrown and the wildfire effects on slope stability have diminished. |
| has surficial material -Colluvium-Usually | (Bovis and Jakob, 1999) | Debris flows are common is areas with easily erodible material. |
| has surficial material -Morainal Material (Till)-Always | (Bovis and Jakob, 1999) | Debris flows are common is areas with easily erodible material. |
| has stream order -1-Always | (Hungr et al., 2014) | Debris flows occur periodically on established paths, usually gullies and first- or second- order streams. |
| has stream order -2-Always | (Hungr et al., 2014) | Debris flows occur periodically on established paths, usually gullies and first- or second- order streams. |

| | | |
|---|---|---|
| has stream order -3-rarely | (Hungr et al., 2014) | Debris flows occur periodically on established paths, usually gullies and first- or second- order streams. |
| has stream order -4-rarely | (Hungr et al., 2014) | Debris flows occur periodically on established paths, usually gullies and first- or second- order streams |
| has stream order -5-rarely | (Hungr et al., 2014) | Debris flows occur periodically on established paths, usually gullies and first- or second- order streams |
| has been logged within years ->20 years-sometimes | (Jackson Jr, 2019) | By 20 year since logging, trees have regrown and the roots are anchoring the soil again. |
| has geomorph process -Debris Flow-always | (Bovis and Jakob, 1999; Wilford et al., 2004) | Melton ratio (number that takes into account relief and area of a watershed) and watershed length allows discrimination of debris flow, debris flood, and flood prone fans. |
| has landslide type-debris flow-Always | (Hungr et al., 2014) | Debris flows occur periodically on established path. Determining the frequency of event is a non-trivial task, but the fact that someone mapped a debris flow in a specific channel, indicates the channel as prone to debris flows events. |
| has landslide type-Fall -usually | (Bovis and Jakob, 1999) | Any landslide types may accumulate debris in a channel that can be then mobilized into a debris flow |
| has landslide type-Flow-usually | (Bovis and Jakob, 1999) | Any landslide types may accumulate debris in a channel that can be then mobilized into a debris flow |
| has landslide type-Slide-usually | (Bovis and Jakob, 1999) | Any landslide types may accumulate debris in a channel that can be then mobilized into a debris flow |
| has landslide type-Spread-usually | (Bovis and Jakob, 1999) | Any landslide types may accumulate debris in a channel that can be then mobilized into a debris flow |
| has landslide type-Topple-usually | (Bovis and Jakob, 1999) | Any landslide types may accumulate debris in a channel that can be then mobilized into a debris flow |

| | | |
|---|---|---|
| has landslide type-Slope deformation-usually | (Bovis and Jakob, 1999) | Any landslide types may accumulate debris in a channel that can be then mobilized into a debris flow |
| has slope -Very steep-always | (Hungr et al., 2014) | Debris flows occur periodically on established paths, usually gullies and first- or second- order streams |
| has slope -Steep-always | (Hungr et al., 2014) | Debris flows occur periodically on established paths, usually gullies and first- or second- order streams |
| has slope -moderately steep-usually | (Hungr et al., 2014) | Debris flows occur periodically on established paths, usually gullies and first- or second- order streams |
| has slope -moderate-usually | (Hungr et al., 2014) | Debris flows occur periodically on established paths, usually gullies and first- or second- order streams |
| has slope -gentle-rarely | (Hungr et al., 2014) | Debris flows occur periodically on established paths, usually gullies and first- or second- order streams |
| has slope -plain-never | (Hungr et al., 2014) | Debris flows occur periodically on established paths, usually gullies and first- or second- order streams |
| has surficial form -cliff-always | (Howes and Kenk, 1997) | Cliffs indicate steep terrains where sediments may be mobilized as debris flows. |
| has surficial form -cones-always | (Howes and Kenk, 1997) | Cones store sediments that may be remobilized into debris flow. |
| has water -permafrost-always | (Hungr et al., 2014) | Permafrost degradation can destabilize sediments |
| has texture -blocks-always | (Howes and Kenk, 1997) | The presence of blocks can be indicator of landslide processes |
| has texture -rubble-always | (Howes and Kenk, 1997) | The presence of rubble is an indicator of landslide processes. |

**Table E2.** Slides in rock model https://italy.minervageo.com/the-roberti-slides-in-rock-model/

| Instance Property-Value-Frequency | Model Definition Source | Comments |
|---|---|---|
| has geomorph process - GeneralPeriglacialProcesses-always | (Evans and Clague, 1994) | Landslides are common in periglacial environment, especially under changing climatic conditions. |
| has geomorph process - ErosionalProcess-always | (Guzzetti et al., 2012) | Active erosional processes are possible indicator of landslide activity, as landslides occur where landslides have occurred before. |
| has geomorph process - MassMovement-always | (Guzzetti et al., 2012) | Active mass movement processes are possible indicator of landslide activity, as landslides occur where landslides have occurred before. |
| has slope -Very Steep-always | (Hungr et al., 2014) | Very Steep slopes are prone to slides |
| has slope -Steep-always | (Hungr et al., 2014) | Steep slopes are prone to slides |
| has slope -Moderately Steep-usually | (Hungr et al., 2014) | Moderately steep slopes are prone to slides |
| has slope -Moderate-sometimes | (Hungr et al., 2014) | Moderate slopes may be prone to slides |
| has slope -Gentle-never | (Hungr et al., 2014) | Gentle slopes are rarely prone to slides |
| has slope -Plain-never | (Hungr et al., 2014) | Plain slopes are rarely prone to slides. |
| has surficial material -Bedrock-ususally | (Hungr et al., 2014) | 'bedrock' mapped as surficial material indicates the presence of cliffs and bluffs, possibility prone to rock slides. |
| has surficial material -Weathered Bedrock-always | (Hungr et al., 2014) | Weather bedrock is more likely to fail than fresh bedrock. |
| has weather threshold -Extreme Weather-always | (Friele, 2012; Segoni et al., 2018) | Landslides can be triggered by intense rainfall (Segoni et al., 2018) or snowmelt. Rainfall threshold for this study are derived from (Friele, 2012). |

| | | |
|---|---|---|
| has weather threshold -Severe Weather-usually | (Friele, 2012; Segoni et al., 2018) | Landslides can be triggered by intense rainfall (Segoni et al., 2018) or snowmelt. Rainfall threshold for this study are derived from (Friele, 2012). |
| has weather threshold -Mild Weather-rarely | (Friele, 2012; Segoni et al., 2018) | Landslides can be triggered by intense rainfall (Segoni et al., 2018) or snowmelt. Rainfall threshold for this study are derived from (Friele, 2012). |
| has weather threshold -Moderate Weather-sometimes | (Friele, 2012; Segoni et al., 2018) | Landslides can be triggered by intense rainfall (Segoni et al., 2018) or snowmelt. Rainfall threshold for this study are derived from (Friele, 2012). |
| has land use -Alpine-always | (Evans and Clague, 1994) | Landslides are common in the Alpine zone, especially under changing climatic conditions |
| has land use -SubAlpineAvalancheChutes-always | (Hungr et al., 2014) | Rock slides can occur in gullies that are also avalanche tracks |
| has stream order -1-always | (Strahler, 1957) | Stream erosion can affect slope stability |
| has stream order -2-always | (Strahler, 1957) | Stream erosion can affect slope stability |
| has stream order -3-always | (Strahler, 1957) | Stream erosion can affect slope stability |
| has stream order -4-usually | (Strahler, 1957) | Stream erosion can affect slope stability |
| has stream order -5-sometimes | (Strahler, 1957) | Stream erosion can affect slope stability |
| has transport line -Road Resource-usually | (Jackson Jr, 2019) | Logging roads are the greatest aggravating factor for landslide activity as compared to undisturbed slopes. |
| has transport line -Road Unclassified Or Unknown-usually | (Jackson Jr, 2019) | Roads are an aggravating factor for landslide activity as compared to undisturbed slopes. |
| has transport line -Trail-usually | (Jackson Jr, 2019) | Roads are an aggravating factor for landslide activity as compared to undisturbed slopes. |

| | | |
|---|---|---|
| has transport line -Road Recreation Demographic-sometimes | (Jackson Jr, 2019) | Roads are an aggravating factor for landslide activity as compared to undisturbed slopes. |
| has water -Permafrost-always | (Jackson Jr, 2019) | Landslides are common in periglacial environment, especially under changing climatic conditions. |
| has bed rock -metamorphic rock-always | (Hungr et al., 2014) | Foliated metamorphic rocks are usually weak and prone to failure. |
| Has CORINE land cover-Glacier and perpetual snow-always | (Evans and Clague, 1994) | Landslides are common in the Alpine zone, especially under changing climatic conditions. |
| has CORINE land cover-Bare rocks-always | (Hungr et al., 2014) | Rock outcrops can be steep and prone to landslides |
| has CORINE land cover-Road and rail networks and associated lands-always | (Jackson Jr, 2019) | Roads and rail increase landslide activity as they are a break in slope where water can accumulate |
| has fault -Any Fault-always | (Reichenbach et al., 2018) | Faults are indicator of weak rocks, and the presence of faults is one of the main parameters considered in landslide susceptibility mapping. |
| has landslide type-Rock Fall-usually | (Guzzetti et al., 2012) | Landslides are more likely to occur on slopes or valleys that have experienced landslides before. |

| | | |
|---|---|---|
| has landslide type-Rock Slope Spread-usually | (Guzzetti et al., 2012) | Landslides are more likely to occur on slopes or valleys that have experienced landslides before |
| has landslide type-Rock topples-usually | (Guzzetti et al., 2012) | Landslides are more likely to occur on slopes or valleys that have experienced landslides before |
| has landslide type-Slides in Rock-always | (Guzzetti et al., 2012) | Landslides are more likely to occur on slopes or valleys that have experienced landslides before |
| has landslide type-Slides in soil-sometimes | (Guzzetti et al., 2012) | Note that location must also be considered. In essence, where there is soil, it is less likely that there will be steep slopes, but soil slides are a sign of an unstable slope, and therefore are not explicitly negatively correlated to rock slides |
| has landslide type-Slope deformation in rock-usually | (Guzzetti et al., 2012) | Landslides are more likely to occur on slopes or valleys that have experienced landslides before |
| has landslide type-Flows in soil-sometimes | (Guzzetti et al., 2012) | Where there is soil, it is less likely that there will be steep slopes, and rock slides. But soil slides are a sign of an unstable slope, and therefore are not explicitly negatively correlated to rock slides |
| has landslide type-Soil Fall-sometimes | (Guzzetti et al., 2012) | Where there is soil, it is less likely that there will be steep slopes, and rock slides. But soil slides are a sign of an unstable slope, and therefore are not explicitly negatively correlated to rock slides |

| | | |
|---|---|---|
| has landslide type-Slope deformation in soil-sometimes | (Guzzetti et al., 2012) | Where there is soil, it is less likely that there will be steep slopes, and rock slides. But soil slides are a sign of an unstable slope, and therefore are not explicitly negatively correlated to rock slides |
| has landslide type-Soil Topple-sometimes | (Guzzetti et al., 2012) | Where there is soil, it is less likely that there will be steep slopes, and rock slides. But soil slides are a sign of an unstable slope, and therefore are not explicitly negatively correlated to rock slides |
| has surficial form -cliff-always | (Hungr et al., 2014) | Cliffs can generate rock slides |
| has texture-rubble-Always | (Howes and Kenk, 1997) | The presence of blocks can be indicator of landslide processes |
| has texture-blocks-Always | (Howes and Kenk, 1997) | The presence of rubble is an indicator of landslide processes. |
| has surficial form -Cones-Always | (Howes and Kenk, 1997) | Cones may be formed by rock slide debris, hence they can be considered an indicator of rockslide activity |

**Table E3.** Slides in soil model https://italy.minervageo.com/slides-in-soil/

| Instance Property-Value-Frequency | Model Definition Source | Comments |
|---|---|---|
| has surficial material -Morainal Material (Till)-always | (Jackson Jr et al., 2008) | Soil slides can be generated when morainal material fails from a slope |
| has surficial material -Bedrock-sometimes | (Jackson Jr et al., 2008) | There may be some soil even when 'bedrock' has been mapped as principal surficial material |
| has surficial material -Colluvium-always | (Jackson Jr et al., 2008) | Soil slides can be generated when colluvium has been mapped as principal surficial material |
| has geomorph process -ErosionalProcess-always | (Guzzetti et al., 2012) | Active erosional processes are possible indicator of landslide activity, as landslides occur where landslides have occurred before. |
| has geomorph process -MassMovement-always | (Guzzetti et al., 2012) | Active mass movement processes are possible indicator of landslide activity, as landslides occur where landslides have occurred before. |
| has slope -Plain-rarely | (Hungr et al., 2014) | Soil slides rarely occur on plain slopes. |
| has slope -Gentle-rarely | (Hungr et al., 2014) | Soil slides rarely occur on plain slopes. |
| has slope -Moderate-usually | (Hungr et al., 2014) | Soil slides usually occur on moderate slopes. |
| has slope -Moderately Steep-usually | (Hungr et al., 2014) | Soil slides usually occur on moderate steep slopes. |
| has slope -Steep-rarely | (Hungr et al., 2014) | Soil slides rarely occur on moderate steep slopes, because usually there is not much soil on steep slopes. |
| has slope -Very Steep-never | (Hungr et al., 2014) | Soil slides rarely occur on steep slopes, because usually there is not much soil on steep slopes. |
| has land use -Alpine-never | (Hungr et al., 2014) | Soil slides rarely occur in the Alpine zone, because usually there is not much soil there. |

| | | |
|---|---|---|
| has land use - SubAlpineAvalancheChutes-usually | (Hungr et al., 2014) | Soil slides can occur in the gullies that are also avalanche tracks. |
| has stream order -1-always | (Strahler, 1957) | Stream erosion can cause soil slides |
| has stream order -2-always | (Strahler, 1957) | Stream erosion can cause soil slides |
| has stream order -3-usually | (Strahler, 1957) | Stream erosion can cause soil slides |
| has stream order -4-usually | (Strahler, 1957) | Stream erosion can cause soil slides |
| has stream order -5-sometimes | (Strahler, 1957) | Large stream erosion may cause soil slides |
| has transport line -Trail Skid-always | (Jackson Jr, 2019) | Trail skid are aggravating factor for landslide activity as compared to undisturbed slopes |
| has transport line -Trail-sometimes | (Jackson Jr, 2019) | Trails are an aggravating factor for landslide activity as compared to undisturbed slopes |
| has transport line -Road Resource-always | (Jackson Jr, 2019) | Logging roads are the greatest aggravating factor for landslide activity as compared to undisturbed slopes. |
| has transport line -Road Unclassified Or Unknown-always | (Jackson Jr, 2019) | Roads are an aggravating factor for landslide activity as compared to undisturbed slopes. |
| has transport line -Highway-rarely | (Jackson Jr, 2019) | Roads are an aggravating factor for landslide activity as compared to undisturbed slopes. |
| has transport line -Road Recreation Demographic-sometimes | (Jackson Jr, 2019) | Roads are an aggravating factor for landslide activity as compared to undisturbed slopes. |
| has thickness -Blanket-always | (Jackson Jr et al., 2008) | Soil Slides can occur when there is enough soil that can be mobilized on a slope. |
| has thickness -Mantle of Variable Thickness-usually | (Jackson Jr et al., 2008) | Soil Slides can occur when there is enough soil that can be mobilized on a slope. |
| has thickness -Veneer-sometimes | (Jackson Jr et al., 2008) | Soil Slides can occur when there is enough soil that can be mobilized on a slope. |

| | | |
|---|---|---|
| has thickness -Thin Veneer-rarely | (Jackson Jr et al., 2008) | Soil Slides can occur when there is enough soil that can be mobilized on a slope. |
| has rainfall -Extreme Rainfall-always | (Friele, 2012; Segoni et al., 2018) | Landslides can be triggered by intense rainfall (Segoni et al., 2018) or snowmelt. Rainfall threshold for this study are derived from (Friele, 2012).. |
| has rainfall -Severe Rainfall-usually | (Friele, 2012; Segoni et al., 2018) | Landslides can be triggered by intense rainfall (Segoni et al., 2018) or snowmelt. Rainfall threshold for this study are derived from (Friele, 2012). |
| has rainfall -Moderate Rainfall-sometimes | (Friele, 2012; Segoni et al., 2018) | Landslides can be triggered by intense rainfall (Segoni et al., 2018) or snowmelt. Rainfall threshold for this study are derived from (Friele, 2012). |
| has rainfall -Mild Rainfall-rarely | (Friele, 2012; Segoni et al., 2018) | Landslides can be triggered by intense rainfall (Segoni et al., 2018) or snowmelt. Rainfall threshold for this study are derived from (Friele, 2012). |
| has bed rock -metamorphic rock-always | (Bovis and Jakob, 1999) | Metamorphic foliated rocks have usually weak geotechnical properties. Basin underlain by these weak rocks are likely to experience more landslides compared to basin underlain by stronger lithologies. |
| has texture -blocks-always | (Howes and Kenk, 1997) | The presence of block can be indicator of mass movement processes |
| has texture -rubble-always | (Howes and Kenk, 1997) | The presence of rubble is an indicator of mass movement processes. |
| has been logged within years ->20 years-sometimes | (Jackson Jr, 2019) | By 20 year since logging, trees have regrown and the roots are anchoring the soil again |
| has been logged within years -10-20 years-usually | (Jackson Jr, 2019) | Landslides are likely by 10 to 20 years after tree harvesting as new trees are starting to provide anchoring effect with their roots on the slope. |
| has been logged within years -5-10 years-always | (Jackson Jr, 2019) | Landslides are extremely likely by 5 to 10 years after tree harvesting. Most of tree roots have died, and new trees are too small to provide anchoring effect with their roots on the slope. |

| | | |
|---|---|---|
| has been logged within years -0-5 years-usually | (Jackson Jr, 2019) | Landslides are likely by 0 to 5 years after tree harvesting as the trees are dead but some roots are still providing anchoring effect on the slope. |
| has fire within years ->20 years-sometimes | (Jackson Jr, 2019) | After 20 year since a wildfire, trees have regrown and the wildfire effects on slope stability have diminished. |
| has fire within years -10-20 years-sometimes | (Jackson Jr, 2019) | Landslides are likely between 10 to 20 years after a wildfire. The roots have lost anchoring effect and the new trees are still too small to support the slope. |
| has fire within years -0-2 years-always | (Jackson Jr, 2019) | Landslides are very likely for 2 years after a wildfire. Water cannot infiltrate, runoff and erosion increase as the soil becomes water repellent and loses cohesion because of the fire heat |
| has fire within years -3-5 years-usually | (Jackson Jr, 2019) | Landslides are likely between 3 to 5 years after a wildfire. The water-repellent soil horizon degrades but the roots of dead trees are starting to rot and they do not support the slope with their anchoring effect anymore. |
| has fault -Any Fault-always | (Reichenbach et al., 2018) | The presence of fault is an important factor to determine landslide susceptibility |
| has fire within years -5-10 years-always | (Jackson Jr, 2019) | Landslides are very likely between 5 to 10 years after a wildfire. Roots of dead trees decay, and they are not supporting the soil anymore as for the case of tree harvesting logging. |

| | | |
|---|---|---|
| has landslide type-Slides in soil-always | (Guzzetti et al., 2012) | Landslides are more likely to occur on slopes or valleys that have experienced landslides before. |
| has landslide type-Fall in rock-sometimes | (Guzzetti et al., 2012) | Where there is rock, it is less likely that there will be soil slides rather than landslides in rock. But landslides in rock are a sign of an unstable slope, and therefore are not explicitly negatively correlated to soil slides |
| has landslide type-Rock topples-sometimes | (Guzzetti et al., 2012) | Where there is rock, it is less likely that there will be soil slides rather than landslides in rock. But landslides in rock are a sign of an unstable slope, and therefore are not explicitly negatively correlated to soil slides |
| has landslide type-Flows in rock-sometimes | (Guzzetti et al., 2012) | Where there is rock, it is less likely that there will be soil slides rather than landslides in rock. But landslides in rock are a sign of an unstable slope, and therefore are not explicitly negatively correlated to soil slides |
| has landslide type-slides in rock-sometimes | (Guzzetti et al., 2012) | Where there is rock, it is less likely that there will be soil slides rather than landslides in rock. But landslides in rock are a sign of an unstable slope, and therefore are not explicitly negatively correlated to soil slides |

| | | |
|---|---|---|
| has landslide type-Slope deformation in rock -sometimes | (Guzzetti et al., 2012) | Where there is rock, it is less likely that there will be soil slides rather than landslides in rock. But landslides in rock are a sign of an unstable slope, and therefore are not explicitly negatively correlated to soil slides |
| has landslide type-Spread in rock-sometimes | (Guzzetti et al., 2012) | Where there is rock, it is less likely that there will be soil slides rather than landslides in rock. But landslides in rock are a sign of an unstable slope, and therefore are not explicitly negatively correlated to soil slides |

*Author contributions.*

- Gioachino Roberti, Jakob McGregor, Clinton Smyth and David Poole wrote the paper

- Gioachino Roberti conceptually designed the susceptibility schema and landslide extension, the expert-based landslide models and expanded the geohazard ontology.

- Jakob McGregor implemented the INSPIRE schema and code list extension and designed the web map application.

- David Poole and Clinton Smyth designed the qualitative probabilistic method used to calculate susceptibility.

- Sharon Lam and Blake Boyko implemented and maintain the web map.

- Victoria Wang implemented and maintained the geohazard ontology.

- Bryan Barnhart and Chris Ahern implemented the qualitative probabilistic algorithm.

- Stephen Richards supported the semantic implementations and edited the manuscript.

- David Bigelow helped in the redaction of the manuscript, reviewed the landslide models and the code list extensions.

*Competing interests.* The authors declare that they have no conflict of interest

*Acknowledgements.* This project was first presented at the Helsinki 2019 INSPIRE data challenge, and won the first prize. The authors would like to acknowledge the conference organizer committee including the National Land Survey of Finland and Ministry of Agriculture, and the Joint Research Centre of the European Commission and Spatineo. The authors would also like to acknowledge M. Alvioli et al., for the
440 availability of the r.slopeunit code, and M. Mergili and S. Pudasaini, for the r.avaflow code, and WeTransform GMBH for the HALE Connect and HALE Studio software licences. And the reviewers I. Marchesini and O. F. Althuwaynee for constructive feedbacks on the manuscript. This research was funded by Minerva Intelligence Inc.

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
