# Peer review of "INSPIRE standards as framework for artificial intelligence applications: a landslide example"

_Natural Hazards and Earth System Sciences, 2020_

## Referee Comment (RC1) · Omar F. Althuwaynee (Referee) · 20 Jun 2020

Reviewer report: nhess-2020-134 "INSPIRE standards as framework for artificial intelligence applications: a landslides example"

In the presented work an application that tries to demonstrate the value of INSPIRE compliance in enhancing the knowledge interoperability in field of Landslide susceptibility analysis. The manuscript introduced, highlighted and discussed valuable information and critical points and current issues in mapping natural hazards using spatial data and AI.

However, the reviewer still need to discuss here some points that still need to be elaborated within the text for clear the confusion with readers, especially those who are on

various level of experience or scientific backgrounds.

General comments:

Abstract: 1- What kind of enhancement the authors refer to rather than standardize the knowledge interoperability using the standard vocabularies, please clarify this. 2- "The use of INSPIRE-standardized vocabularies in ontologies that express scientific models promotes the adoption of the standards across the European Union and beyond" This sentence need enormous work to be carried to fulfill its objective, in your current work, how much or how far did you consider your present work contributes to this end? Taking in mind the various methods of the Risk assessment researcher: Data selection and representation, variable selection and optimization, modeling type physical to datamining...etc. 3- . As matter of fact, any analytical model using spatial data, suffers from uncertainty started with modeling ambiguity, surrogate data, error propagation..etc. in different extend, Therefore, the extension to Natural Risk zone susceptibility schema, need to consider the model accuracy assessment, validation and error quantification of data (spatial and aspatial) and used model

1 introduction L27 P2 : "EU countries are aligning and .... Europe (Cho and Crompvoets, 2019)." Most of the high quality sensors collected data and field works supported by scientists located in Europe, thus, Please mention the main rebuttals why the application still limited in literature

L43 P2: More standards are still missing need to be mentioned here, like: 1- Standards for input data volume concerning the study area extent and landslide type. 2- Standard for outcomes accuracy concerning the type/quality/amount and extent of reserch details coverage.

L45 P2 : "Deep learning techniques....such as JPEG, or WAV)" Deep learning still not widely applied in Hazard and vulnerability mapping. The authors may mentioned the most common models in the literature instead like physical or datamining statistical models.

[Figure]

L210 to 221 P 10: too general information, please discuss in more details the suscep­tibility modeling like in light of using machine learning (classification/clustering..etc) or physical model of slope structure analysis.

L275 P 17: "Quantification of this value has yet to be made, but calculations.. Lom­bardia Region, Italy, alone." Please extend this paragraph by mentioning some details or statistics.

L285 to L315 P 18: Can we assume that, the mentioned limitations, were the reason behind the scare mentioned in L28,29 P2 " EU countries are aligning and serving INSPIRE data at a slow pace...are available across Europe"?

L323, 324 L19: " However, in current geological assessments, expert judgment is still widely applied (e.g., Association of Professional Engineers and Geoscientists of British Columbia, 2010)," Please add recent references to support this statement.

Specific comments: L55 to L60 P3: Not clear, please re-write. L125 P3: "in geohazard mapping to produce landslide susceptibility maps (Jackson Jr et al., 2008)" Can you add more recent studies. Figure 2. Please mention one study for each sub-category Figure 3 and 5. . Please use multiple layout as the figure is not readable on A4 paper mode.

---

## Referee Comment (RC2) · Ivan Marchesini (Referee) · 22 Jun 2020

**1   Introduction**

Reading this manuscript was interesting and stimulating. The article deals with the problem of landslide susceptibility mapping by combining different aspects, ranging from (i) the definition of a landslides classification compliant with the INSPIRE Directive, (ii) the definition of a multi-hierarchical model for the same classification, (iii) the definition of an INSPIRE scheme for landslide susceptibility and (iv) the definition of an expert-based method for the generation of maps of susceptibility to specific types of landslides in areas where inventories are scarce. In this sense the paper is adequate to the journal also because it presents innovative concepts which are adequate to in-

ternational standards. Despite this I think that interventions are needed to improve the paper and make it suitable to be published.

**2   General comments**

The manuscript makes extensive use of the term artificial intelligence (also in the title). However, the type of approach used is not the classical application of artificial intelligence expected from the literature. I think it is important to clarify why this terminology is used (using the literature) or alternatively to modify the title and the text focusing more on ontology and taxonomy. In the latter case some of the titles used for the different sections of the manuscript should be changed.

In the Method section, I think that there is a need of an introductory sub-sections which briefly introduce the flowchart of the method, even using a figure. I suggest something like: (I) defining taxonomies, (ii) defining expert-based rules, (iii) performing matching, (iv) deriving the susceptibility map. As a consequence of the flow-chart introduction the subsequent sub-sections could use a title which is compliant with the flowchart content.

Sub-section 3.1.1 describe the creation of the NH classification code list for landslides. It was used for defining the expert-based rules. I wonder if this sub-section should be moved in the method section. Moreover it could be helpful to describe how the flat representation of your classification differs from the classification of Hungr et al. (2014)

The method used for building the susceptibility map is based on the definition of the rules and setting of the matching scores. I wonder if these scores can have a huge impact on the final calculation of the map. I think that a discussion about how the scores are defined and which is the effect of changing those values is needed. Looking at the matching score table it seems that unmatch means -10. Is this something that should be added in the text?

[Figure]

The set of rules used for defining the expert-based model are important and should be visible. May be I'm wrong but I can't find a table or a web address where those rules can be observed. I think that for a reader it could be very helpful to have access to those set of rules.

In the conclusions you stated that in this study you present some landslide susceptibility maps. I would rather say that that you present a method for building a landslide susceptibility map using taxonomy and predictors/covariates and that this method is useful where accurate landslides inventories are not available.

About figures and tables: figure 5 is not clear. I would transform it to a table and I would let the user to go and see it in the website putting a link in the caption.

**3  Specific comments**

Row 45: I would remove the reference to jpeg or wav.

Row 50: I think that the concept of entities and relationships should be defined a priori or some literature should be cited.

Rows 52 -54: This sentence is not immediately clear. An example could help the reader.

Row 81: In my opinion the approach is based on the domain-expert reasoning, since rules are defined a priory. Then it mimics these rules.

At rows 104-105 the definitions of the model is too cryptic. I would add some examples to make clear to the reader that, if I understand correctly, a model is a set of rules defined a priori by the expert and based on the properties of the entities defined in the taxonomy.

Row 110: semantic triple format and semantic network. Please cite a reference or

define them.

Row 111: "revered" or "referred" ?

Rows 135-136: I would use "channels" and "steep channels" in place of "streams"

Row 160: About the stream line vector layer. I suppose that only the segments in in the mountain valleys were used. But what about the starting point of these segments? This is relevant since debris flows can be triggered also in the upper part of the watersheds where channels may not have been delineated. You can discus this point for completeness.

Row 167: is r.avaflow suitable also for slides in rock?

Rows 224-226: please use an example to describe what you have done to align the data to INSPIRE standards and explain why the same was not done for the other datasets (lakes, watersheds, etc).

Table 2: please explain how you have used the IFFI database in your model. I suppose it was used for helping to define the expert-based rules.

Table 3: how the Watersheds, Railroad and Road layers have been used in the model?

Rows 231-234: what about the slope map? Was the map expressed in terms of classes of slope and not in degrees or percentage?

Row 245: 99,9th percentile of the slope units susceptibility values. I suppose. Please specify it.

Row 275: please remove brackets

---

## Author Response (AR1)

**INSPIRE standards as framework for artificial intelligence applications: a  landslide example**

Gioachino Roberti[1], Jacob McGregor[1], Sharon Lam[1], David Bigelow[1], Blake Boyko[1], Chris Ahern[1], Victoria Wang[1], Bryan Barnhart[1], Clinton Smyth[1], David Poole[1,2], and Stephen Richard[1]

[1]Minerva Intelligence Inc., 301 – 850 West Hastings Street, Vancouver, British Columbia, Canada V6C 1E1
[2]Computer Science department, University of British Columbia, Vancouver, V6T 1Z4, Canada

**Correspondence:** Gioachino Roberti (groberti@minervaintelligence.com)

**Abstract.** This study presents a landslide susceptibility map using an artificial intelligence (AI) approach  based on standards set by the INSPIRE framework.  INSPIRE is a European Union Spatial Data Infrastructure (SDI) initiative to standardize spatial data across borders to ensure interoperability for management of cross-border infrastructure and environmental issues.  However, despite the theoretical effectiveness of the SDI,  few real-world applications make use of INSPIRE standards. In this study, we show how INSPIRE standards enhance the interoperability of geospatial data, and enable deeper knowledge development for their interpretation and explainability in AI applications. We designed an ontology of landslides, embedded with INSPIRE vocabularies and then aligned geology, stream network and land cover data sets covering the Veneto region of Italy to the standards. INSPIRE was formally extended to include an extensive landslide type code list, a landslide size code list and the concept of landslide susceptibility to describe map application inputs and outputs. Using the terms in the ontology, we defined conceptual scientific models of  areas likely to generate different type of landslides as well as map polygons representing  the land surface. Both landslide models and map polygons were encoded as semantic networks and, by qualitative probabilistic comparison between the two, a similarity score was assigned. The score was then used as a proxy for landslide susceptibility and displayed in web map application. The use of INSPIRE-standardized vocabularies in ontologies that express scientific models promotes the adoption of the standards across the European Union and  globally. Further, this application facilitates  explaining the generated results. We conclude that public and private organisations, within and outside the European Union, can enhance the value of their data by bringing them into INSPIRE-compliance for use in AI applications.

*Copyright statement.* This work is distributed under the Creative Commons Attribution NonCommercial-ShareAlike 4.0 International
https://creativecommons.org/licenses/by-nc-sa/4.0/

[revised manuscript text omitted]

**List of all relevant changes made in the manuscript**

- Added method figure and re-arranged the methods accordingly, as per suggestion of Reviewer 2
- Modified UML figure showing only the extension implemented in this project
- Deleted former Figure 5 and replaced it with Tables 4 and 5.
- Edited introduction to clarify the AI approach
- Edited the methods to clarify models, and scoring system
- Edited results and discussions
- Clarified in the discussion the main contribution of this work, which are:
    - the susceptibility schema extension
    - code list extensions
    - an example application making use of using ontologies embedded with standardized terminology, leveraging the INSPIRE spatial data infrastructure
- Added links to the model and tables showing all of the models in the appendices.
- General text clean-up

**Point-by-point response to the Editor**

Comments to the Author:
dear authors,

after checking the referees' reports, the discussion section and your submitted manuscript, we believe that your paper has potential interest for the journal readers but it is still in a form which is not acceptable by the journal, due to some issues that are quite important.
For this reason, we will reconsider a new, revised submission should you be ready to incorporate some major changes, as requested by the two referees.

Dear Editor, thank you for considering the manuscript for submission to the journal, we have addressed all the points raised by the reviewers and edited the manuscript accordingly.

In particular, I feel that some of their requests are particularly reasonable and, therefore, mandatory to bring the manuscript up to the journal standards. I refer to:

1. the need to clarify the methodology with respect to the (many) existing standards existing in landslide hazard studies

We re-organized the methodology and results, and a summary of landslide susceptibility mapping approaches is provided in section 1.4. A full review of landslide hazard methods is beyond the scope of this project, and as noted, the community is far from agreeing on unique standard.

Section 1.4 reads: "Landslide susceptibility is defined as the relative spatial probability of occurrence for a landslide based on the intrinsic properties of a site (SafeLand, 2011). The concept of susceptibility differs from hazard in that the temporal probability of occurrence, the triggering factors, and the magnitude of the event are not considered in the definition of a susceptibility map (SafeLand,2011; Van Den Eeckhaut and Hervás, 2012). To produce landslide susceptibility maps, three approaches are usually applied: statistical, physical, and expert-based (SafeLand, 2011). Statistical methods rely on the analysis of landslide databases and their relation to landscape properties (see review by Reichenbach et al., 2018); physical methods calculate the limit equilibrium between failure resisting forces and driving forces in slopes (e.g., Baum et al., 2008); and expert-based methods rely on expert opinion and

the assumption that influencing factors are known and are specified in the models (Dai et al., 2002). The AI approach used in this study is an example of the expert-based approach, as the models follow rules that represent the reasoning process of a landslide-expert, providing semi-quantitative susceptibility map"

2. the need to clarify, homogenize and standardize the taxonomy, the ontology and the terminology used (including the way you refer to the landslide object inside a classification method which is rather different from the ones widely accepted

I do not understand this comment. The landslide taxonomy refers to the updated Varnes classification, the ontology uses CGI UIGS rock taxonomy, British Columbia terrain mapping standards, Corinne Land cover and Strahler stream order and other standardized classifications. The susceptibility schema extension is drafted from the Safeland 2011 standards.

3. the need to detail some parts of the methodology by adding figures and documentation (e.g. flow chart and expert rules used)

We addressed this point by re-organizing the manuscript as suggested by Reviewer 2.

4. the need to try to quantify the impact of the different possible choices of rules and settings on the final result

While we certainly agree that our chosen "rules" and "settings" should be scrutinized, we emphasize that the final result of the paper is not the assessments of landslide susceptibility. Instead, the final result is the susceptibility extension in INSPIRE, as well as the development of code lists and a framework within which landslide susceptibility data can be encoded. The ontology-based landslide susceptibility assessment is intended to be used as an example of the benefits of applying the developed INSPIRE landslide framework, rather than an exhaustive assessment of landslide susceptibility. For this reason, we believe that a sensitivity analysis would be beyond the scope of this paper.

5. the authors should be more careful in stating quite uncertain and very challenging objectives, such as those of unifying EU INSPIRE standards in AI intelligence applications and terminology since those standards are still debated and non-existing at the moment. Moreover, they cannot be defined by force since they require a general agreement among the many institutions and consortiums working on landslides and related risks. The task is overwhelmingly complex and the authors do not seem to clarify what is the contribution of their work towards this general objective

Our contributions are discussed in section 4 , and we have clarified some of the language used in this section to more clearly articulate our specific contribution, which includes the following three accomplishments:
- the susceptibility schema extension
- code list extensions
- an example of the benefits of applying INSPIRE standards in the form of ontologically-driven landslide susceptibility assessments

See our reply to Reviewer 1:
In this study we start from showing how to modify INSPIRE to make it possible to use it for landslide-specific applications. By suggesting a landslide-specific schema and code list extensions, we set the ground for INSPIRE-compliant landslide susceptibility studies. Other organizations can build on top of these extensions and future landslide susceptibility applications can be compared as they formally refer to the same data structure and semantics. Note that we do not force any specific data and modeling variable selection, nor modeling approach for landslide susceptibility/hazard/risk method. Such an effort is beyond the scope to this paper and, to some extent, already addressed by the SafeLand project, rather, we provide the data structure and semantics to store and share whichever method has been chosen by the modeler. For example, data selection for calculation of landslide

susceptibility is encompassed in the schema structure under "Influencing Factor" which are "unbounded in multiplicity and can be defined qualitatively or quantitatively", leaving a broad range of possibilities to the modeler.

6. the need to better frame the entire method and results within the existing landslide risk mitigation framework (by also citing the most relevant literature). When the proposed tools would be actually usable and useful? Possibly, where no accurate landslide inventory maps are already available? And under which conditions and pre-requisites?

As noted in the response to (1), we re-organized the methodology and results, and a summary of landslide hazard approaches is provided in section 1.4. We also clarified some of the language that specifies when the ontological approach can (or should) be used, in section 4.3.

Section 1.4: "Landslide susceptibility is defined as the relative spatial probability of occurrence for a landslide based on the intrinsic properties of a site (SafeLand, 2011). The concept of susceptibility differs from hazard in that the temporal probability of occurrence, the triggering factors, and the magnitude of the event are not considered in the definition of a susceptibility map (SafeLand,2011; Van Den Eeckhaut and Hervás, 2012). To produce landslide susceptibility maps, three approaches are usually applied: statistical, physical, and expert-based (SafeLand, 2011). Statistical methods rely on the analysis of landslide databases and their relation to landscape properties (see review by Reichenbach et al., 2018); physical methods calculate the limit equilibrium between failure resisting forces and driving forces in slopes (e.g., Baum et al., 2008); and expert-based methods rely on expert opinion and the assumption that influencing factors are known and are specified in the models (Dai et al., 2002). The AI approach used in this study is an example of the expert-based approach, as the models follow rules that represent the reasoning process of a landslide-expert, providing semi-quantitative susceptibility map"

Added in section 4.3: "The main goal of this paper is not to present the semantic matching approach, but to show an example on how to modify INSPIRE to make it possible to use it for landslide-specific applications. By suggesting these landslide-specific schema and code list extensions, we set the ground for INSPIRE-compliant landslide susceptibility studies. Other organizations can build on top of these extensions and future landslide susceptibility applications can be compared as they formally refer to the same data structure and semantics. Note that we do not force any specific data and modeling variable selection, nor modeling approach for landslide susceptibility/hazard/risk method. Such an effort is beyond the scope to this paper and, to some extent, already addressed by the SafeLand project (e.g., SafeLand, 2011) rather, we provide the data structure and semantics to store and share whichever method has been chosen by the modeler. For example, data selection for calculation of landslide susceptibility is encompassed in the schema structure under "Influencing Factor" which are "unbounded in multiplicity and can be defined qualitatively or quantitatively", leaving broad range of possibilities to the modeler. Regarding the data quality, it is discussed in the Natural Risk Zone schema and they refer to ISO standards (INSPIRE Thematic Working Group Natural Risk Zones, 2013). However, we recognize that specific code list (semantics) dealing with data quality and model uncertainty are missing. We hope that the INSPIRE thematic group will address this point.

Please also answer to all the remaining minor issues and specific comments highlighted in the reviewer's reports, one by one, by modifying the manuscript where required and directly replying to all remarks and comments on a separate document to attach to the new modified manuscript.

The replies to all the reviewers' comments are below, in this document.

I am confident that, after that, the second review round might be expeditious and more successful.

Thank you.

**Point-by-point response Reviewer 1.**

In the presented work an application that tries to demonstrate the value of INSPIRE
compliance in enhancing the knowledge interoperability in field of Landslide susceptibility
analysis. The manuscript introduced, highlighted and discussed valuable information
and critical points and current issues in mapping natural hazards using spatial
data and AI. However, the reviewer still need to discuss here some points that still need to be elaborated
within the text for clear the confusion with readers, especially those who are on
various level of experience or scientific backgrounds.

GR: Dear reviewer, Thank you for the insightful revision and useful comments. Our replies are below your
comments and we edited the text to address your observations when needed.

General comments:

Abstract:

1- What kind of enhancement the authors refer to rather than standardize
the knowledge interoperability using the standard vocabularies, please clarify this.

GR: We enhance knowledge transfer and interoperability of data and data analytics as they are based on the same
data structure and semantic standards.

2-"The use of INSPIRE-standardized vocabularies in ontologies that express scientific _
models promotes the adoption of the standards across the European Union and beyond"
This sentence need enormous work to be carried to fulfill its objective, in your
current work, how much or how far did you consider your present work contributes
to this end? Taking in mind the various methods of the Risk assessment researcher:
Data selection and representation, variable selection and optimization, modeling type
physical to datamining...etc.

GR: In this study we start from showing how to modify INSPIRE to make it possible to use it for landslide-specific
applications. By suggesting these landslide-specific schema and code list extensions, we set the ground for
INSPIRE-compliant landslide susceptibility studies. Other organizations can build on top of these extensions and
future landslide susceptibility application can be compared as they formally refer to the same data structure and
semantics. Note that we do not force any specific "Data selection and representation, variable selection and
optimization, modeling type physical to datamining...etc" for landslide susceptibility/hazard/risk method, rather,
we provide the data structure and semantics to store and share whichever method has been chosen by the
modeler. For example, data selection is encompassed in the schema structure under "Influencing Factor" which
are "unbounded in multiplicity and can be defined qualitatively or quantitatively", leaving broad range of
possibilities to the modeler.

To clarify this point, we added this paragraph in section 4.3: "The main goal of this paper is not to present the
semantic matching approach, but to show an example on how to modify INSPIRE to make it possible to use it for
landslide-specific applications. By suggesting these landslide-specific schema and code list extensions, we set the
ground for INSPIRE-compliant landslide susceptibility studies. Other organizations can build on top of these
extensions and future landslide susceptibility applications can be compared as they formally refer to the same data
structure and semantics. Note that we do not force any specific data and modeling variable selection, nor
modeling approach for landslide susceptibility/hazard/risk method. Such an effort is beyond the scope to this
paper and, to some extent, already addressed by the SafeLand project (e.g., SafeLand, 2011) rather, we provide

the data structure and semantics to store and share whichever method has been chosen by the modeler. For example, data selection for calculation of landslide susceptibility is encompassed in the schema structure under "Influencing Factor" which are "unbounded in multiplicity and can be defined qualitatively or quantitatively", leaving broad range of possibilities to the modeler. Regarding the data quality, it is discussed in the Natural Risk Zone schema and they refer to ISO standards (INSPIRE Thematic Working Group Natural Risk Zones, 2013). However, we recognize that specific code list (semantics) dealing with data quality and model uncertainty are missing. We hope that the INSPIRE thematic group will address this point.

3- . As matter of fact, any analytical model using spatial data, suffers from uncertainty started with modeling ambiguity, surrogate data, error propagation..etc. in different extend, Therefore, the extension to Natural Risk zone susceptibility schema, need to consider the model accuracy assessment, validation and error quantification of data (spatial and aspatial) and used model.

GR: Data quality standards are discussed in the Natural Risk Zone schema and they refer to ISO standards (Section 7 and 8 in D2.8.III.12 Data Specification on Natural Risk Zones). However, we recognize that specific code list (semantics) dealing with data quality and model uncertainty are missing. We hope that the INSPIRE thematic group will address this point. We briefly mention the importance of models parameters semantics at lines 329-331 "by embedding the ontology concepts related to statistical parameters (e.g. receiving operating curves, confidence intervals) or physical parameters (e. g. friction angles, viscosity), it will be possible for the numerical outputs of quantitative methods to be explained in natural language"

1 introduction

L27 P2 : "EU countries are aligning and .... Europe (Cho and Crompvoets, 2019)." Most of the high quality sensors collected data and field works supported by scientists located in Europe, thus, Please mention the main rebuttals why the application still limited in literature.

GR: In-depth discussion on why INSPIRE is slowly adopted is beyond the scope of this paper. Cho and Crompvoets, (2019) suggest that the slow INPIRE adoption by EU countries may be due to legal and policy issues. Regarding the scientific literature, there are a few cases which make use of INSPIRE, and we discuss them in the paper. Furthermore, INSPIRE is a geospatial framework which is not something commonly discussed in the geological/geomorphological literature.

L43 P2: More standards are still missing need to be mentioned here, like: 1- Standards for input data volume concerning the study area extent and landslide type. 2- Standard for outcomes accuracy concerning the type/quality/amount and extent of reserch details coverage.

GR: This is a general introduction to standards and AI, beyond the field of natural hazards. Regarding the "1- Standards for input data volume concerning the study area extent and landslide type and 2- Standard for outcomes accuracy concerning the type/quality/amount and extent of research details coverage". These are a currently discussed topics in the geological/geomorphological literature, and there is no universal agreement on such standards. This paper focuses on the INSPIRE semantic and data framework standards in which multiple different approaches to landslide susceptibility mapping can fit and can provide interoperable results.

L45 P2 : "Deep learning techniques....such as JPEG, or WAV)" Deep learning still not widely applied in Hazard and vulnerability mapping. The authors may mentioned the most common models in the literature instead like physical or datamining statistical models.

GR: This is a general introduction to standards and AI, beyond the field of natural hazards. The methods to asses landslide susceptibility are discussed in section 1.4 of the paper, but a detailed review of physical and statistical methods for landslide susceptibility mapping is beyond the scope of this paper.

Section 1.4: "Landslide susceptibility is defined as the relative spatial probability of occurrence for a landslide based on the intrinsic properties of a site (SafeLand, 2011). The concept of susceptibility differs from hazard in that the temporal probability of occurrence, the triggering factors, and the magnitude of the event are not considered in the definition of a susceptibility map (SafeLand,2011; Van Den Eeckhaut and Hervás, 2012). To produce landslide susceptibility maps, three approaches are usually applied: statistical, physical, and expert-based (SafeLand, 2011). Statistical methods rely on the analysis of landslide databases and their relation to landscape properties (see review by Reichenbach et al., 2018); physical methods calculate the limit equilibrium between failure resisting forces and driving forces in slopes (e.g., Baum et al., 2008); and expert-based methods rely on expert opinion and the assumption that influencing factors are known and are specified in the models (Dai et al., 2002). The AI approach used in this study is an example of the expert-based approach, as the models follow rules that represent the reasoning process of a landslide-expert, providing semi-quantitative susceptibility map"

L210 to 221 P 10: too general information, please discuss in more details the susceptibility modeling like in light of using machine learning (classification/clustering..etc) or physical model of slope structure analysis.

GR: In this paragraph we present a detailed description of the schema structure and how inputs and outputs of susceptibility modelling can be mapped to this schema. We do not discuss how susceptibility modelling can be done, leaving space to the many possible approaches.  A overview of physical and statistical methods for landslide susceptibility mapping is discussed in section 1.4 of the paper.

Section 1.4: "Landslide susceptibility is defined as the relative spatial probability of occurrence for a landslide based on the intrinsic properties of a site (SafeLand, 2011). The concept of susceptibility differs from hazard in that the temporal probability of occurrence, the triggering factors, and the magnitude of the event are not considered in the definition of a susceptibility map (SafeLand,2011; Van Den Eeckhaut and Hervás, 2012). To produce landslide susceptibility maps, three approaches are usually applied: statistical, physical, and expert-based (SafeLand, 2011). Statistical methods rely on the analysis of landslide databases and their relation to landscape properties (see review by Reichenbach et al., 2018); physical methods calculate the limit equilibrium between failure resisting forces and driving forces in slopes (e.g., Baum et al., 2008); and expert-based methods rely on expert opinion and the assumption that influencing factors are known and are specified in the models (Dai et al., 2002). The AI approach used in this study is an example of the expert-based approach, as the models follow rules that represent the reasoning process of a landslide-expert, providing semi-quantitative susceptibility map"

L275 P 17: "Quantification of this value has yet to be made, but calculations.. Lombardia Region, Italy, alone." Please extend this paragraph by mentioning some details or statistics.

GR: We realized that sentence was incorrect and rephrased: "A comparative study (Craglia and Campagna, 2010) of regional SDI in the context of INSPIRE implementation, showed that inefficient data access and use at the European level results in economic losses in the 100-200 Million Euro annual range. The same study, shows that the regional SDI of Lombardia, Italy, allowed € 3 m/year savings to companies working in Environmental Impact Assessments (EIA), and Strategic Environmental Assessments (SEA). Savings in the same order of magnitude can be expected by adopting INSPIRE standards in the geological hazard assessment domain."

L285 to L315 P 18: Can we assume that, the mentioned limitations, were the reason behind the scare mentioned in L28,29 P2 " EU countries are aligning and serving

INSPIRE data at a slow pace...are available across Europe"?

GR: The mentioned limitations are part of the reasons. Other reasons maybe that is a lot of work to align the data and the return of money and time investments for data transformation is not immediately quantifiable

L323, 324 L19: " However, in current geological assessments, expert judgment is still widely applied (e.g., Association of Professional Engineers and Geoscientists of British Columbia, 2010)," Please add recent references to support this statement.

GR: Unfortunately, guidelines for professional practice are not updated very often. In this context a reference from 2010 is to be considered "recent".  We edited the text and added another reference. Now reads: "However, in current professional geological assessments and geomorphological mapping applications, expert judgment is still widely applied (e.g., Association of Professional Engineers and Geoscientists of British Columbia, 2010, Guzzetti et al., 2012),"

Specific comments: L55 to L60 P3: Not clear, please re-write.

GR: We rephrased: "For example, "Slides in soil" and "Slides in rock" share the same parent concept "Slides" and they are differentiated by the property dealing with the material type, "Soil" and "Rock", which make them uniquely identifiable. Taxonomies based on Aristotelian definitions support multi-hierarchical knowledge networks and can be used by computers to make logical inferences (Poole et al., 2009; Smith, 2003). The term 'multi-hierarchical' implies that there is more than one way to move through a taxonomy to arrive at a particular node or term. For example, the landslide taxonomy can be arranges based on different properties. If the landslide types are firstly arranged based on the type of movement and then based on the type of material, one path within the taxonomy would be: Landslide> slides> slides in rock and slides in soil. Alternatively, if the landslide types are arranged first based on the material type and then on the movement type, the path of the taxonomy would be: Landslide> landslides in rock> slides in rock and flows in rock. Both paths are valid, but they reach the same concept in different ways

L125 P3: "in geohazard mapping to produce landslide susceptibility maps (Jackson Jr et al., 2008)" Can you add more recent studies.

GR: There is no other recent study using this expert-based approach based on ontological matching for landslide susceptibility mapping. The framework presented in this paper can be adopted with any method used to assess landslide susceptibility. In the schema, landslide susceptibility is an element that can be quantitatively or qualitatively defined.

Figure 2. Please mention one study for each sub-category

GR: I do not understand this comment. Do you mean a reference for each landslide type? They are mainly from Hungr et al 2014.. details on the properties used for the classification are in   Appendix B - Properties used for the landslide classification

Figure 3 and 5. Please use multiple layout as the figure is not readable on A4 paper mode.

GR: We modified figure 3, showing only the extension done for this project, and we deleted Figure 5 and replace it with Tables 4 and 5. In the tables captions there is the link to the actual match report table from the webmap application

**Point-by-point response to Reviewer 2.**

1 Introduction

Reading this manuscript was interesting and stimulating. The article deals with the problem of landslide susceptibility mapping by combining different aspects, ranging from (i) the definition of a landslides classification compliant with the INSPIRE Directive, (ii) the definition of a multi-hierarchical model for the same classification, (iii) the definition of an INSPIRE scheme for landslide susceptibility and (iv) the definition of an expert-based method for the generation of maps of susceptibility to specific types of landslides in areas where inventories are scarce. In this sense the paper is adequate to the journal also because it presents innovative concepts which are adequate to international standards. Despite this I think that interventions are needed to improve the paper and make it suitable to be published.

GR: Dear reviewer, thank you for your careful revision and your insightful comments. We have addressed point by point your comments and updated the text accordingly.

2 General comments

The manuscript makes extensive use of the term artificial intelligence (also in the title). However, the type of approach used is not the classical application of artificial intelligence expected from the literature. I think it is important to clarify why this terminology is used (using the literature) or alternatively to modify the title and the text focusing more on ontology and taxonomy. In the latter case some of the titles used for the different sections of the manuscript should be changed.

GR: As stated in line 34, "Artificial intelligence" is "the synthesis and analysis of computational agents that act intelligently" (Poole and Mackworth, 2017). This definition encompasses a broad range of methods and algorithms including, but not limited to, machine learning. I understand that commonly the term artificial intelligence is used as synonym for machine learning, which in turn is a term used to talk about various statistical methods. We rephrased the paragraph.

We rephrased the paragraph to clarify this point: "The term Artificial Intelligence is commoly used to indicate only the machine learning part of the field, especially in the landslide literature (e.g., Dieu and Gjermundsen, 2020). In this paper we use the term in its broader connotation, which includes also the ontological method used in this paper. See below for the description of the method and definition of ontologies"

In the Method section, I think that there is a need of an introductory sub-sections which briefly introduce the flowchart of the method, even using a figure. I suggest something like: (I) defining taxonomies, (ii) defining expert-based rules, (iii) performing matching, (iv) deriving the susceptibility map. As a consequence of the flow-chart introduction the subsequent sub-sections could use a title which is compliant with the flowchart content.

GR: Good point, thank you. We have added a flowchart figure and updated the methods adding an intro. We also have rearranged Methods and Results to remove some of the repetition and add details in the webmap final results.

Sub-section 3.1.1 describe the creation of the NH classification code list for landslides. It was used for defining the expert-based rules. I wonder if this sub-section should be moved in the method section. Moreover it could be helpful to describe how the flat representation of your classification differs from the classification of Hungr et al. (2014)

GR: We reword methods and results to clarify this point: "The Natural Hazard Classification code list extension for landslides considers material type and failure movement, splitting the tree, first on type of movement, and then on type of material, following Hungr et al. (2014) (Figure 3). Other properties, such as: water content, depth of failure, rate of movement, loading state, channelized state, and failure plane geometry (see Appendix B) are used to describe the individual landslide types, as the unique combination of these properties allows for unambiguous classification in an Aristotelian taxonomy. We used these properties because, even if not shown in the final taxonomic tree, they are explicitly applied in the wordy description of landslides type by Hungr et al. (2014).

The method used for building the susceptibility map is based on the definition of the rules and setting of the matching scores. I wonder if these scores can have a huge impact on the final calculation of the map. I think that a discussion about how the scores are defined and which is the effect of changing those values is needed. Looking at the matching score table it seems that unmatch means -10. Is this something that should be added in the text?

GR: You are right, these scores determine the final calculation of the map. While we certainly agree that our chosen "rules", "settings" and "scores" should be scrutinized, we emphasize that the final result of the paper is not the assessments of landslide susceptibility. Instead, the final result is the susceptibility extension in INSPIRE, as well as the development of code lists and a framework within which landslide susceptibility data can be encoded. The ontology-based landslide susceptibility assessment is intended to be used as an example of the benefits of applying the developed INSPIRE landslide framework, rather than an exhaustive assessment of landslide susceptibility. For this reason, we believe that a sensitivity analysis would be beyond the scope of this paper

That said, a detailed description of the scores is provided in lines 115-125; they are a measure of surprise that uses order of magnitude numbers to distinguish qualitative measures. To better explain and address the – 10, we added in the text at line 125: "In this study, an Exact match or an AKO exact match of a property with frequency "always" scores 10000, "usually" scores 9000, "sometimes" scores 1000, "rarely" scores "100" and "never" scores -10000; unmatched attributes are awarded -10 points. These scores are an arbitrary representation of degree of surprise that uses order of magnitude numbers to distinguish qualitative measures.

For more extensive review we refer Smyth and Poole 2004:

- *always a proposition is "always" true, means that you are very surprised if it is false. All experience leads you to believe that it is always true, but you are leaving open the possibility that it isn't true*
- *usually a proposition is "usually" true means you are some- what surprised if it is false.*
- *sometimes a proposition that is "sometimes" true, means you wouldn't be surprised if it is true or false.*
- *rarely a proposition is "rarely" true means you are some- what surprised if it is true.*
- *never a proposition is "never" true, means that you are very surprised if it is true.*

*In terms of the kappa calculus, we choose numbers α> 0 and β> 0 so that:*

*• always p means κ(¬p) = α and κ(p) = 0. Thus α is the measure of surprise that p is false.*

*• usually p means κ(¬p) = β and κ(p) = 0. Thus β is the measure of surprise that p is false. The relative surprises means that β< α.*

*• sometimes p means κ(¬p) = 0 and κ(p) = 0. We are not surprised if p is true or not.*

*• rarely p means κ(¬p) = 0 and κ(p) = β. Thus β is the measure of surprise that p is true. Note that "rarely" is the dual of "usually."*

*•never p means κ(¬p) = 0 and κ(p) = α. Thus α is the measure of surprise that p is true. Note that "never" is the dual of "always."*

*Note that these qualitative uncertainties are only the input values (i.e., as part of the models); on output we give a numerical score (both a raw score as well as a percent of the best match). This finite scale is not adequate to describe the level of matches. In our applications we have used β = 1000 as the value for being very surprised. (The only significant as the value for being somewhat surprised and α = 10000 feature of the values is the 10-fold ratios between them; 10 "somewhat surprised" is equal to one "very surprised").*

*Different expert models would lead to different susceptibility maps.*

The set of rules used for defining the expert-based model are important and should be visible. Maybe I'm wrong but I can't find a table or a web address where those rules can be observed. I think that for a reader it could be very helpful to have access to those set of rules.

GR Links to the models are in appendix C. We also added them in the text in section 2.2.1.

We added in section 2.2.1: "These landslide models are intended to be proof-of-concept of models that can be used in the semantic approach proposed in this paper. In particular, some of the properties used in the models are drafted from literature analysis of logging-related landslides in British Columbia, Canada"

In the conclusions you stated that in this study you present some landslide susceptibility maps. I would rather say that that you present a method for building a landslide susceptibility map using taxonomy and predictors/covariates and that this method is useful where accurate landslides inventories are not available.

GR: Yes, we agree. We rephrased the first sentences of the conclusions "This study presents an AI-based method to produce landslide susceptibility maps using an ontology and standardized taxonomies within the framework provided by the INSPIRE Natural Risk Zone theme. This method does not need an accurate landslides inventory to make predictions. We produced susceptibility maps for debris flow, slides in soil and slides in rock for the province of Veneto, Italy."

About figures and tables: figure 5 is not clear. I would transform it to a table and I would let the user to go and see it in the website putting a link in the caption.

GR: Good point. We made the figure into two tables (4 and 5), added the link in the tables' captions, and updated the text

Specific comments

Row 45: I would remove the reference to jpeg or wav.

GR: We rephrased to: "commonly specified in data storage standards such as JPEG, or WAV"

Row 50: I think that the concept of entities and relationships should be defined a priori or some literature should be cited.

GR: We expanded the sentence and added a reference: "In particular, an ontology defines the vocabulary for individuals and relationships within a knowledge domain. Individuals may be concrete entities (e.g. a rock), or abstract concepts, (e.g. numbers); relationships are properties that describe how individuals are connected. Typical examples of relationships include: is-a-kind-of, is-part-of, is-superclass-of, has-some-property; the ontology also defines axioms controlling the use of the vocabulary for logical and thematic consistency Poole2017"

Rows 52 -54: This sentence is not immediately clear. An example could help the reader.

GR: We added an example: "For example, "Slides in soil" and "Slides in rock" share the same parent concept "Slides" and they are differentiated by the property dealing with the material type, "Soil" and "Rock", which make them uniquely identifiable. Taxonomies based on Aristotelian definitions support multi-hierarchical knowledge networks and can be used by computers to make logical inferences (Poole et al., 2009; Smith, 2003)"

Row 81: In my opinion the approach is based on the domain-expert reasoning, since rules are defined a priory. Then it mimics these rules.

GR: This is correct. The method applies rules that are set to follow expert reasoning. We rephrased to better explain: "The AI approach used in this study is an example of the expert-based approach, as the models follow rules that represent the reasoning process of a landslide-expert, providing semi-quantitative susceptibility map"

At rows 104-105 the definitions of the model is too cryptic. I would add some examples to make clear to the reader that, if I understand correctly, a model is a set of rules defined a priori by the expert and based on the properties of the entities defined in the

taxonomy.

GR: You are correct, we rephrased and added links to the expert models in section 2.2.1 and tables in appendix C.

"A model is a set of rules defined a priori by an expert, based on scientific literature, making use of the entities and properties defined in the ontology. These models aim to represent expert conceptualized descriptions of a given phenomenon or entity (e.g. landslide susceptibility)."

Row 110: semantic triple format and semantic network. Please cite a reference or define them.

GR: We rephrased: "Semantic networks are a graph representation of knowledge where nodes are concepts and edges are the semantic relation between concepts (Shapiro, 1992)"

Row 111: "revered" or "referred" ?

GR: Sorry for the typo, we fixed it

Rows 135-136: I would use "channels" and "steep channels" in place of "streams"

GR: Ok, text is updated

Row 160: About the stream line vector layer. I suppose that only the segments in in the mountain valleys were used. But what about the starting point of these segments? This is relevant since debris flows can be triggered also in the upper part of the watersheds where channels may not have been delineated. You can discus this point for completeness.

GR: Stream line vector layer is from the Veneto Region geoportal. The dataset is not great: many channels, especially in the initiation zones, are not represented. But the idea was to use as much as possible data already available, rather than making new data. We want to provide a framework for interoperable landslide susceptibility mapping: many other methods, better data etc could be used to assess landslide susceptibility, but still can be delivered using this schema and code list extension

Row 167: is r.avaflow suitable also for slides in rock?

GR: We think so: r.avaflow allows to model fluid and solid fraction separately, you can play around with the parameters and recreate the runout of wide range of landside types, including slides in rock, assuming that the rock mass disintegrate and starts to behave as a flow-like landslide.

Rows 224-226: please use an example to describe what you have done to align the data to INSPIRE standards and explain why the same was not done for the other datasets (lakes, watersheds, etc).

GR: We rephrased the paragraph and added a figure showing the Hale Studio user interface: "For this study, we used open access datasets from the Veneto Region Geoportal and other sources (Table 2 and 3). Aligning all input datasets was beyond the scope of this project. We did, however, want to show the value of INSPIRE-aligned data and therefore aligned stream network, CORINE land cover, bedrock geology, and the Italian Landslide Inventory (IFFI) (Table 2) to INSPIRE using Hale Studio (WeTransform, 2008). Figure 5 shows how different tools in Hale Studio are used to align properties from the source dataset to the target dataset. For example, the field "eta" – "Age" in Italian, of the original Veneto dataset, was directly mapped to four different INSPIRE fields: the olderNamedAge.href and title and the youngerNamedAge.href and title. Note that olderNamedAge.href youngerNamedAge.href are hyperlinks to the code list value id and the title is the actual code list term from the GeochronologicEraValue code list. This alignment is done with many classification methods, including: Groovy Scripts, formatted strings and assign-alignment tools. For further explanation on term alignments, refer to the documentation of Hale Studio (WeTransform, 2008). Datasets used that were not compliant with INSPIRE include: lakes, watersheds, permafrost, fire, slope angle, faults, soil, roads and railways (Table 3).

Table 2: please explain how you have used the IFFI database in your model. I suppose it was used for helping to define the expert-based rules.

GR: Yes, it is correct. Expert-based model states that: Slides in Soil - has landslide type - slides in soil - always. As landslides are more likely to occur on slopes or valleys that have experienced landslides before. See Appendix C.

Table 3: how the Watersheds, Railroad and Road layers have been used in the model?

GR: Watersheds have been used to with Melton Ratio to classify catchment as debris flow, debris flood and flood prone

Roads and rail roads have been used by assuming that roadcut and railroad cut affect slope stability, when compared to undisturbed slopes. See appendix C.

Rows 231-234: what about the slope map? Was the map expressed in terms of classes of slope and not in degrees or percentage?

GR: Yes, slope classes are based on degrees. The matching systems adopted in this paper requires discretized data, which can be a benefit, as numbers (e.g. 37°) are harder than words (e.g. steep) to understand by non-technical people

Row 245: 99,9th percentile of the slope units susceptibility values. I suppose. Please specify it.

GR: Yes, correct, we rephrased to: "99,9th percentile score (i. e. susceptibility values) of instances (slope units and stream buffer polygons) for each landslide type"

Row 275: please remove brackets

GR: That sentence was actually incorrect. We rephrased: "A comparative study (Craglia and Campagna, 2010) of regional SDI in the context of INSPIRE implementation, showed that inefficient data access and use at the European level results in economic losses in the 100-200 Million Euro annual range. The same study, shows that the regional SDI of Lombardia, Italy, allowed € 3 m/year savings to companies working in Environmental Impact Assessments (EIA), and Strategic Environmental Assessments (SEA). Savings in the same order of magnitude can be expected by adopting INSPIRE standards in the geological hazard assessment domain."